# Separable roles of the DNA damage response kinase Mec1$^{ATR}$ and its activator Rad24$^{RAD17}$ during meiotic recombination

**Margaret R. Crawford**[1,2]**, Jon A. Harper**[1]**, Tim J. Cooper**[1]**, Marie-Claude Marsolier-Kergoat**[3,4]**, Bertrand Llorente**[5]**, Matthew J. Neale**[1]*

**1** Genome Damage and Stability Centre, School of Life Sciences, University of Sussex, United Kingdom, **2** Francis Crick Institute, London, United Kingdom, **3** Institute for Integrative Biology of the Cell (I2BC), CEA, CNRS, Univ. Paris-Sud, Université Paris-Saclay, Gif-sur-Yvette, France, **4** UMR7206 Eco-Anthropology and Ethno-Biology, CNRS-MNHN-University Paris Diderot, Musée de l'Homme, Paris, France, **5** Cancer Research Centre of Marseille, CNRS, INSERM U1068, Institut Paoli-Calmettes, Aix-Marseille Université UM105, Marseille, France

* m.neale@sussex.ac.uk

**Data Availability Statement:** Octad and tetrad sequences are publicly available at the NCBI Sequence Read Archive (accession numbers

## Abstract

During meiosis, programmed DNA double-strand breaks (DSBs) are formed by the topo-isomerase-like enzyme, Spo11, activating the DNA damage response (DDR) kinase Mec1$^{ATR}$ via the checkpoint clamp loader, Rad24$^{RAD17}$. At single loci, loss of Mec1 and Rad24 activity alters DSB formation and recombination outcome, but their genome-wide roles have not been examined in detail. Here, we utilise two strategies—deletion of the mis-match repair protein, Msh2, and control of meiotic prophase length via regulation of the Ndt80 transcription factor—to help characterise the roles Mec1 and Rad24 play in meiotic recombination by enabling genome-wide mapping of meiotic progeny. In line with previous studies, we observe severely impacted spore viability and a reduction in the frequency of recombination upon deletion of *RAD24*—driven by a shortened prophase. By contrast, loss of Mec1 function increases recombination frequency, consistent with its role in DSB *trans*-interference, and has less effect on spore viability. Despite these differences, complex multi-chromatid events initiated by closely spaced DSBs—rare in wild-type cells—occur more frequently in the absence of either Rad24 or Mec1, suggesting a loss of spatial regulation at the level of DSB formation in both. Mec1 and Rad24 also have important roles in the spatial regulation of crossovers (COs). Upon loss of either Mec1 or Rad24, CO distributions become more random—suggesting reductions in the global manifestation of interference. Such effects are similar to, but less extreme than, the phenotype of 'ZMM' mutants such as *zip3*Δ, and may be driven by reductions in the proportion of interfering COs. Collectively, in addition to shared roles in CO regulation, our results highlight separable roles for Rad24 as a pro-CO factor, and for Mec1 as a regulator of recombination frequency, the loss of which helps to suppress any broader defects in CO regulation caused by abrogation of the DDR.

SRP151982, SRP152540, SRP152953). Analysis code can be found at https://github.com/Neale-Lab/OctadRecombinationMapping. Recombination events images are publicly available at https://figshare.com/s/89100901b905324fc50f and https://doi.org/10.25377/sussex.27188226.v1.

**Funding:** M.R.C, T.J.C, and M.J.N were supported by a European Research Council Consolidator Grant (#311336), the Biotechnology and Biological Sciences Research Council (#BB/M010279/1) and the Wellcome Trust (#200843/Z/16/Z). J.H. and M.J.N. are supported by the Wellcome Trust (#225852/Z/22/Z). B.L. lab was funded by the ANR-13-BSV6-0012-01 grant from the Agence Nationale de la Recherche and a grant from the Fondation ARC pour la Recherche sur le Cancer (SFI20121205448). The funders had no role in study design, data collection and analysis, decision to publish, or preparation of the manuscript.

**Competing interests:** The authors have declared that no competing interests exist.

## Author summary

$Mec1^{ATR}$, and its associated activator, $Rad24^{RAD17}$, are components of the evolutionarily conserved DNA damage response (DDR)—a surveillance network that monitors the integrity of the genome within vegetative cells. The DDR has also been co-opted to regulate processes specific to meiosis—a unique nuclear division in which the induction and repair of DNA breaks via recombination is crucial to create genetically diverse haploid gametes. As such, a detailed molecular understanding of how meiotic recombination is regulated is relevant to our broader understanding of evolution, gametogenesis, and fertility. Here, we reveal that, despite their functional relationship as activator-effector, $Rad24^{RAD17}$ and $Mec1^{ATR}$ have overlapping, yet distinct, functions in meiosis. Notably, whilst both factors suppress the formation of complex recombination events and help COs to be more evenly distributed across the genome, Rad24 acts as a pro-CO factor and is of greater importance for spore viability. By contrast, Mec1 is predominantly required to restrain the total amount of recombination that may take place during any given meiosis. Collectively our findings deepen our understanding of the roles of the meiotic DDR, suggesting both shared and also specialist modes of action.

## Introduction

Meiosis is a specialized form of cell division that produces haploid cells for sexual reproduction. Integral to meiosis is the process of genetic recombination, which is initiated by programmed DNA double-strand breaks catalysed by the topoisomerase II-like enzyme, Spo11 [1]. Meiotic recombination is monitored by the DNA damage response (DDR) in a similar manner to DNA lesions arising within vegetative cells, likely due to the potentially dangerous nature of DSBs. In the generalised pathway, DNA damage leads to the activation of $Tel1^{ATM}$ (Ataxia Telangiectasia-Mutated) and $Mec1^{ATR}$ (Rad3-related) checkpoint kinases (Human orthologues are indicated with superscript text), via the Mre11-Rad50-$Xrs2^{NBS1}$-complex and $Rad24^{RAD17}$ (reviewed in [2]). $Rad24^{RAD17}$ is the loader of the $Ddc1^{RAD9}$-$Rad17^{RAD1}$-$Mec3^{HUS1}$ ("9-1-1") checkpoint clamp, that binds to the ssDNA/dsDNA junctions that arise following resection of DNA ends [3]. $Tel1^{ATM}$ and $Mec1^{ATR}$ modulate downstream targets via $Rad53^{CHK2}$ [4], causing cell cycle arrest and the modulation of transcription [5]. Notably, in mammalian cells, RAD17 and ATR have both overlapping and distinct functions in the DDR. RAD17 is required for the recruitment of RAD9 after DNA damage, a function that it performs even in the absence of ATR [6]. Conversely, ATR can localize to sites of DNA damage independently of RAD17, and can also be activated by other factors such as RPA-coated ssDNA [7], suggesting separable and complementary roles for ATR and RAD17.

In meiosis, Spo11-DSBs initiate homologous recombination and are therefore essential for the generation of crossovers (COs) between homologous chromosomes. In most organisms, including mammals and *S. cerevisiae*, the organism utilised in the work presented here, such recombination-dependent interactions facilitate homologue pairing during leptotene-zygotene, full alignment and connection via the synaptonemal complex at pachytene, and subsequent reductional chromosome segregation at the meiosis I nuclear division [8]. In *S. cerevisiae* meiosis, similar to the vegetative DDR pathway, the 9-1-1 clamp complex, its loader Rad24, and the Mre11-Rad50-Xrs2 complex act as damage sensors [9], with ssDNA, produced by the resection of Spo11-DSBs, activating Mec1 [10]. By sensing ongoing recombination activity and unrepaired DSBs [10,11], Mec1 acts as a molecular rheostat to modulate the progression of meiotic prophase via the Mek1 kinase, a paralogue of Rad53 (CHK2 in mammals)

that regulates the activity of Ndt80 [12,13], the transcription factor required for exit from meiotic prophase [14,15]. Due to this transient checkpoint activation, Mec1 is able to prolong the stage during which Spo11-DSB formation can occur [12,13,16], and both Mec1 and Rad24 promote CO formation and suppress ectopic (non-allelic) recombination [17–19]. The interaction between Mec1 and the 9-1-1 complex also contributes to other Mec1 functions such as phosphorylation of the meiosis-specific chromosome axis protein, Hop1 (HORMAD1/2 in mammals [20]), which is important for the maintenance of homologue bias and chiasma formation [21,22]. In mammals, ATR localises to meiotic chromosomes [23] and is a key regulator of meiotic events. ATR deletion in male mice causes fragmentation of the chromosome axis, and ATR is required for synapsis and loading of recombinases RAD51 and DMC1 at DSB sites [24,25]. In *Drosophila*, loss of Mei-41, the fly ATR orthologue, not only abrogates checkpoint signalling [26], but also alters the frequency and spatial distribution of COs [27]. In *Arabidopsis thaliana*, although *atr* mutants are fully fertile [28], ATR promotes meiotic recombination by regulating the deposition of DMC1 at DSBs [29].

Observations in *S. cerevisiae* indicate that Spo11 DSBs do not occur independently, but are instead subject to interference among the four chromatids (reviewed in [30]). This interference occurs in *cis* (adjacent, on the same chromatid [31]), and *trans* (between chromatids [32]). While both Mec1 and Tel1 are involved in DSB interference and the global suppression of Spo11 activity [31–33]—roles that appear conserved in other organisms [26,34]—a direct role for Rad24 is unclear [31]. CO events also display interference (reviewed in [35]), a process dependent upon the 'ZMM'-family of proteins (reviewed in [36]). Rad24 is necessary for the efficient loading of ZMM proteins to meiotic chromosomes, and interacts physically with Zip3, independently of Mec1 [37], suggesting that Rad24, but not Mec1, may promote interfering CO formation. Interestingly however, genetic measurements on three chromosomes indicate that both Rad24 and Mec1 are important for CO interference [38], suggesting that any separable role in CO formation is distinct from a shared role in promoting interference.

To further establish the roles of Mec1 and Rad24 in the regulation of meiotic recombination, and to determine whether previously inferred roles can be extended genome-wide, we mapped meiotic recombination patterns at high resolution across the *S. cerevisiae* genome in *rad24Δ* and *mec1-mn* ('meiotic null') backgrounds, revealing both similar and distinct roles for Mec1 and Rad24 in the regulation of meiotic DSB repair.

## Results

### Reduced spore viability in *mec1* and *rad24 mutants* can be alleviated by extending meiotic prophase or by deleting *MSH2*

To generate a genome-wide picture of meiotic recombination in wild type, *mec1* and *rad24* mutants, *Saccharomyces cerevisiae* SK1xS288c hybrid diploids (~65,000 SNPs, ~4,000 high confidence INDELs, ~0.57% divergence) were sporulated, the resulting tetrad of spores separated, and genomic DNA sequenced at an average depth of 44x. In strains with a deletion of the mismatch repair (MMR) protein, Msh2, additional heteroduplex (hDNA) tract information can be retrieved by allowing spores to undergo one round of mitotic division and separating them again to form an octad (post-meiotic segregation) prior to sequencing (**Fig 1A**; as described [39–41]). Sequences were aligned against SK1 and S288c reference genomes, and polymorphism information was used to identify regions of recombination using a pipeline (**S1 Fig** and **Methods**) developed in prior studies [39–41].

To avoid the influence of Mec1 inactivation during premeiotic growth, we used a conditional *MEC1* allele (P<sub>CLB2</sub>*MEC1*, hereafter referred to as *mec1-mn* for 'meiotic null') that was previously established in non-hybrid SK1 strains where Mec1 protein became undetectable by

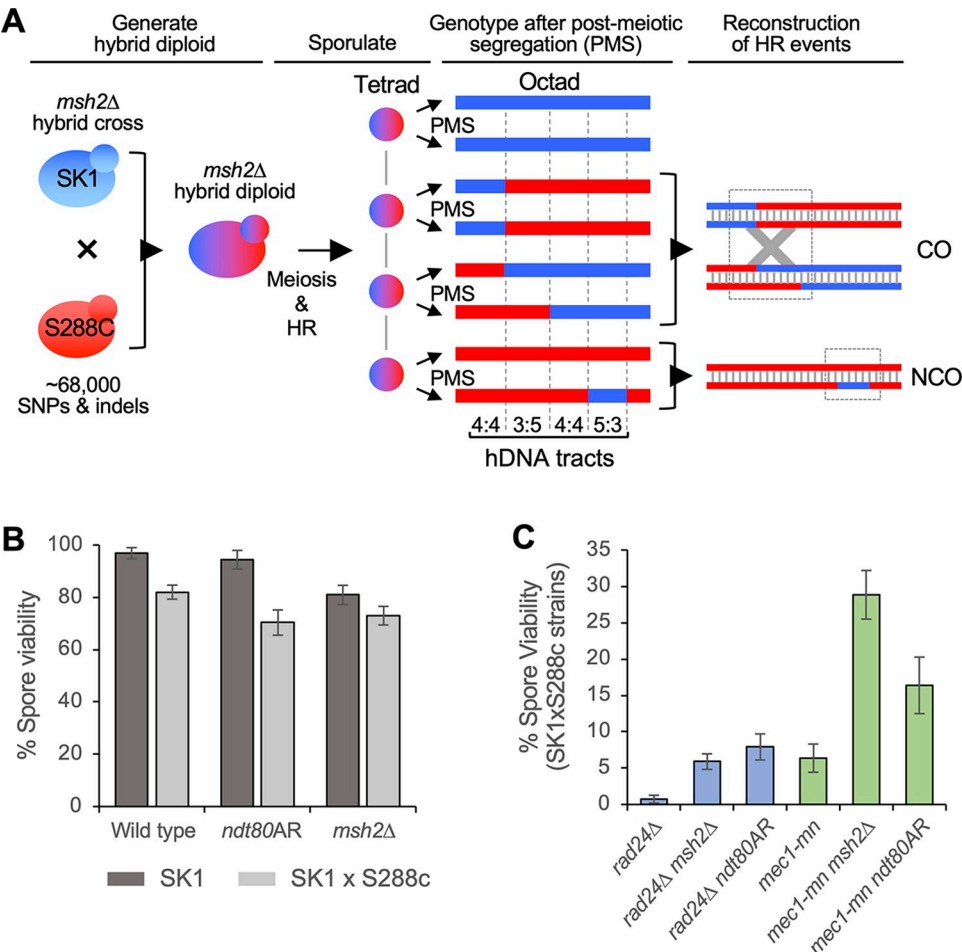

**Fig 1. Overview of recombination analysis strategies. A)** Strategy for whole-genome mapping of HR events in octads after post-meiotic segregation (PMS) of hDNA. For simplicity, a single chromosome is shown containing one CO and one NCO. Haploid cells from the S288c (red) and SK1 (blue) genetic backgrounds are crossed, producing a hybrid diploid containing many polymorphisms. The diploid undergoes meiotic recombination, forming CO and NCO events with associated hDNA tracts. These are left unrepaired in strains with a deletion of the mismatch repair protein Msh2, but are converted or restored in *MSH2* strains. At the conclusion of meiosis, the four chromatids are distributed among four ascospores. To preserve hDNA information, each spore is allowed to undergo one mitotic division, after which the mother and daughter cells are separated to produce eight haploid cell lines, the genomes of which are equivalent to the eight DNA strands involved in the meiotic recombination event. The eight genomes are sequenced to retrieve polymorphism information at each position, allowing precise hDNA reconstruction genome-wide. **B,C)** Spore viability is severely reduced in *rad24Δ* and *mec1-mn*, which is prohibitive to their analysis. To enable sequencing of all four meiotic products, spore viability is increased by prophase extension (8 hours) or *MSH2* deletion. Spore viability comparison for the indicated strains and non/hybrid backgrounds. Error bars indicate 95% confidence limits.

three hours after meiotic induction [16], broadly equivalent to the time at which the molecular steps of recombination begin. Although the degree of Mec1 depletion cannot be assessed in the individual meiotic cells in which we characterise meiotic recombination outcomes, we expect Mec1 protein to be substantially depleted during meiotic prophase.

Accurate determination of recombination patterns by tetrad or octad sequencing requires all spores to survive, hampering the analysis of genotypes with low spore viability (i.e. mutants such as *mec1* and *rad24* that perturb recombination and chromosome segregation). Spore viability is also reduced in hybrid strains compared to pure SK1 or S288c backgrounds (**Fig 1B and S1 Table**); potentially due to the effect sequence divergence has upon recombination

[39,42,43]. Indeed, the already low spore viabilities of *rad24Δ* and *mec1-mn* mutants were exacerbated in the hybrid background **S1 Table**).

To overcome the barrier to analysis caused by low spore viability we explored three strategies. Firstly, we investigated the impact of deleting *SML1 (Suppressor of Mec1 Lethality 1)*, an inhibitor of ribonucleotide reductase that is normally inactivated by Mec1 in response to DNA damage, thereby increasing dNTP pool levels—promoting genetic fidelity and cell viability [44]. Sml1 is not known to have any role in meiosis, nor to affect the viability of a *rad24Δ* mutant. Nevertheless, we found that *sml1Δ* increased the spore viability of *rad24Δ* hybrids ~7 fold (to 4.7%; **S2A Fig**); however, only one four-spore viable tetrad (analysed below) was obtained from 196 tetrad dissections (**S1 Table**), limiting the practicality of this approach.

As a second strategy, we regulated the length of meiotic prophase via an *NDT80* arrest-and-release ('*ndt80AR*') system using an oestradiol-inducible $P_{GAL1}NDT80$ allele [45] that we previously described as a method to increase spore viability in non-hybrid *rad24Δ* mutants [16]. In wild-type SK1 cells, normal prophase I length is ~3–4 hours, with prophase exit coordinated with meiotic DSB repair via decreases in Mec1-dependent activation of Mek1 [12,13,46]. Spore viability was increased by use of the *ndt80AR* system in both *rad24Δ* and *mec1-mn* mutants (**Fig 1C**) and was proportional to the length of time held in prophase (**S2B Fig**). Thus, artificially extending prophase (and/or preventing the early Ndt80 expression that may arise in DDR mutants [14,15]) may provide more time for DDR mutants to undergo and/or complete the productive HR reactions necessary for accurate chromosome segregation at meiosis I [16]. Whilst extension of prophase significantly improved spore viability of *mec1* and *rad24* mutants, it did not greatly impact wild-type spore viability (**Fig 1B**).

The final strategy explored to increase spore viability was deletion of *MSH2*, a mismatch repair (MMR) factor essential for mismatch recognition, in which deletion was previously employed to preserve hDNA tract information for detailed recombination analysis [39,40,47,48]. Abrogating mismatch recognition also reduces heteroduplex rejection by the combined action of MMR factors and the Sgs1-Top3-Rmi1 complex [49], which allows more inter-homologue crossovers to form in a hybrid context and may favour spore viability [39,50]. Indeed, deletion of *MSH2* increased spore viability in hybrid *rad24Δ* and *mec1-mn* strains (**Fig 1C**), despite causing a slight reduction in spore viability in wild type (**Fig 1B**). However, working in the absence of Msh2 presents some limitations since it could change the overall spectrum of events observed in hybrids by allowing inter-homologue recombination where it is prevented under wild-type conditions. Furthermore, Msh2 promotes the formation of complex lesions during mismatch repair [51,52], and also participates in flap cleavage independently of mismatch recognition [49], all of which could impact local and global recombination patterns. Thus, whilst the precise mechanism(s) by which *msh2Δ* increases spore viability in *mec1* and *rad24* mutants are unclear, we favour the idea that it is broadly due to enabling increased efficiency of recombination in hybrid backgrounds where mismatches will be frequently encountered [39].

## Baseline detectable recombination frequencies are increased in *ndt80AR* and *msh2Δ* mutants

Before analysing the effect on recombination of losing Rad24 and Mec1 function, we first assessed the impact of the *ndt80AR* and *msh2Δ* alleles in an otherwise wild-type background. In wild-type cells, an average of 74.5 COs and 30.5 NCOs were detected per meiosis (**Fig 2A** and **S2 Table**), comparable to prior measurements made in this SK1 x S288c hybrid [39]. In all *ndt80AR* strains, oestradiol (to induce Ndt80-dependent exit from prophase) was added 8 hours after transfer to sporulation medium, ~3–4 hours later than the average time at which Ndt80 expression switches on naturally in the wild-type non-hybrid SK1 strain [14].

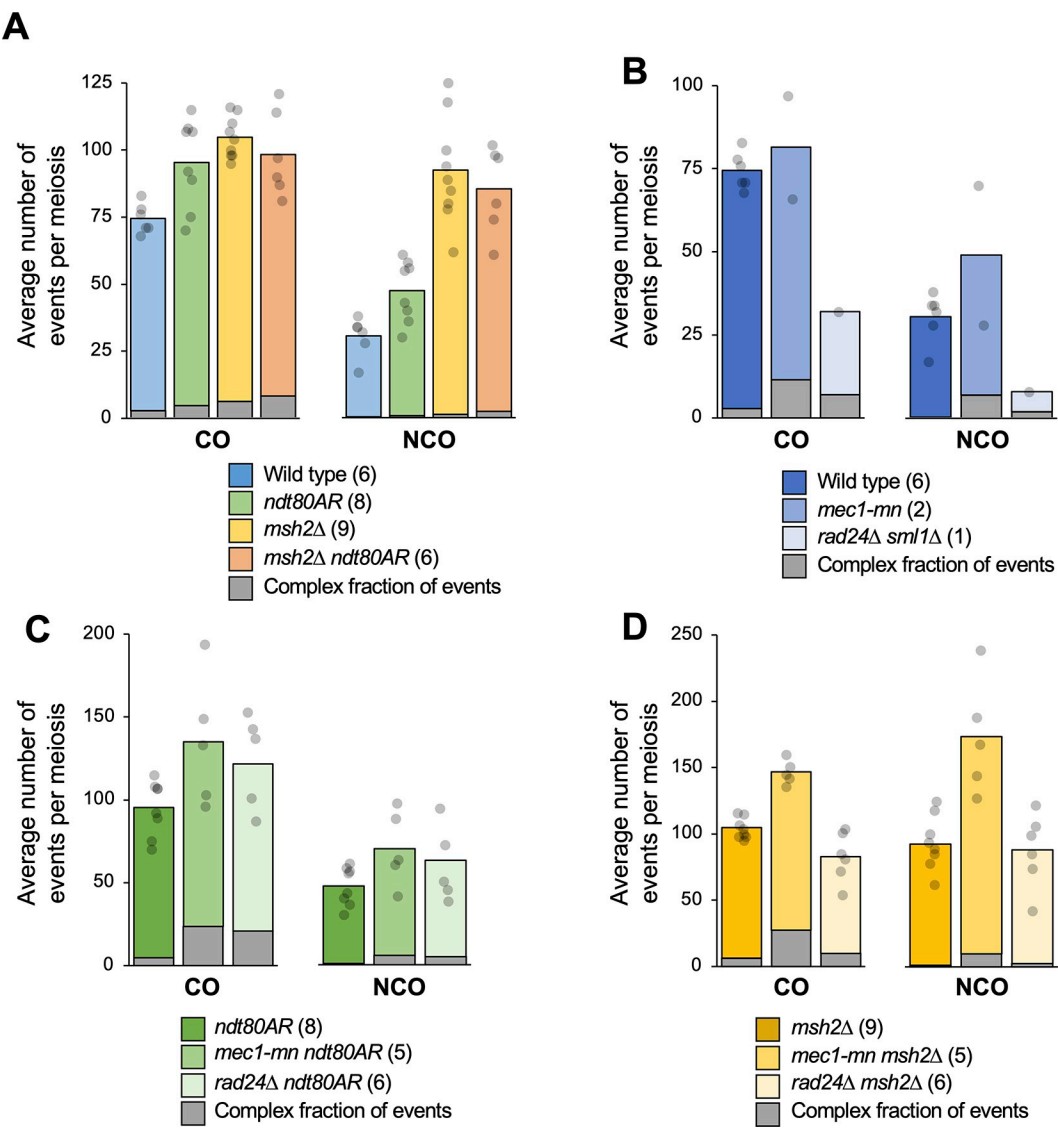

**Fig 2. Meiotic recombination event frequencies per meiosis in the absence of Mec1 or Rad24. A-D)** Recombination event frequencies in control strains (**A**) or in *rad24Δ* and mec1-mn with the genetic background (**B**) wild type or *sml1Δ*; (**C**) *ndt80AR*; (**D**) *msh2Δ*. HR events were computationally scored as a distinct event when separated by at least 1.5 kb of 4:4 marker segregation (nonrecombinant markers). Bar height indicates the mean number of CO and NCO events per meiosis, with individual observations (per meiosis) indicated by grey circles. Number of unique meioses analysed is indicated in parenthesis. The fraction of events involving a greater number of chromatids than expected for a simple HR reaction ("Complex events") are indicated with the grey portion of each bar. For COs, this includes all 3-chromatid and 4-chromatid events. For NCOs, this includes all 3-chromatid and 4-chromatid events, plus those involving 2-sisters. See S1 Table and main text for further details. Tests for significance (Wilcoxon test), as reported in the text, were corrected for multiple testing using the Benjamini-Hochberg method.

Controlling the length of meiotic prophase in this way increased average CO and NCO frequencies ~1.3-fold (CO $P = 0.04$, NCO $P = 0.02$, **Fig 2A**), consistent with the persistent DSB signals observed in terminally-arrested *ndt80Δ* cells [53,54], but also increased the variation observed between individual meioses—particularly for COs—suggesting, perhaps, that individual cells were being held in prophase for differing lengths of time due to asynchronous meiotic induction. By contrast, inactivation of *MSH2* increased CO levels (~1.4-fold, $P<0.01$) more uniformly, but with a greater and rather variable increase (~2.9-fold, $P<0.01$) on

detectable NCOs, as has been observed previously [39]. The effects of *msh2Δ* are likely to be due to a combination of the increased visibility of NCOs in the absence of mismatch restoration, the loss of mechanisms that inhibit recombination at sites of polymorphism in mismatch repair-proficient (*MSH2*+) strains, and the direct role that Msh2 plays during recombination-intermediate metabolism during meiosis [49,51,52]

Whilst no further increase in recombination was observed in the *ndt80AR msh2Δ* double strain (*ndt80AR* $P = 0.846$, *msh2Δ* $P = 0.187$, **Fig 2A**), suggesting that there may be an upper limit to recombination frequency in these backgrounds, we also again observed increased variability in CO levels, presumably caused by the ectopic Ndt80 induction. Interestingly, extending prophase length, but not *MSH2* deletion, skewed recombination towards chromosome ends (**S3A Fig**), an effect that was not observed in centromere-proximal regions (**S3B Fig**). Telomere-proximal effects may be in part due to the nature of end-associated chromosomal regions (EARs), which are both less compacted as measured by Hi-C [55], and retain disproportionately high levels of DSB formation when arrested at pachytene [56]. Notably, although these effects may appear quite subtle, they were observed similarly at all chromosome ends (**S4A and S4B Fig**), suggesting such differences arose due to the extended prophase arrest, and were not driven by any chromosome-specific effects.

## Recombination frequencies are altered in Mec1 and Rad24 mutants to different extents

By increasing spore viability to experimentally tractable levels, the *ndt80AR* and *msh2Δ* alleles provide an avenue to explore genome-wide recombination patterns in DDR mutants for the first time. Nevertheless, it is critical to emphasise that even by adopting such genetic manipulations (*msh2Δ* and *ndt80AR*), spore viability remains extremely low in *rad24Δ* and *mec1-mn* hybrid strains (**Figs 1C** and **S2**), leading to a probable analytical bias towards the less extreme phenotypes that enabled the production of four viable haploid spores. As such, we infer that our descriptions of HR patterns and frequencies presented here should be interpreted cautiously, and encompass only one viewpoint into the potentially complex impact that loss of these critical DDR factors have on meiotic recombination.

Prior analysis indicated that *rad24Δ* mutants display a ~10–20% reduction in global DSB formation compared to controls, thought to be because Mec1-mediated inhibition of Ndt80 is needed to allow sufficient time for wild-type levels of DSBs to form [16]. By contrast, Mec1 has also been reported to suppress DSB formation via the process of *trans*-inhibition [32]. To investigate these and other effects, meiotic recombination was assayed in: one *rad24Δ sml1Δ* and two *mec1-mn* tetrads (**Fig 2B**); five *mec1-mn ndt80AR* and six *rad24Δ ndt80AR* tetrads (**Fig 2C**); and five *mec1-mn msh2Δ* and six *rad24Δ msh2Δ* tetrads (**Fig 2D**). Summarised recombination statistics for each meiosis analysed are presented in **S2 Table**.

The global recombination levels in the two assayed *mec1-mn* tetrads differed from each other by ~1.5-fold (**Fig 2B** and **S2 Table**) both falling close to or outside of the range of CO and NCO frequencies observed in wild type meioses (**Fig 2B**), suggesting that loss of Mec1 activity increases the variability in recombination frequency—at least in those meioses that were successful in generating four viable spores. In both the *msh2Δ* and *ndt80AR* backgrounds, the *mec1-mn* mutant displayed a ~1.4-fold increase in CO formation and a ~1.5–1.9-fold increase in NCO formation compared to the relevant controls (**Fig 2C and 2D**) with greater CO frequency variation observed in *ndt80AR* than in *msh2Δ*, similar to controls.

Interestingly, the enrichment in recombination arising towards chromosome ends observed within *ndt80AR* strains was substantially reduced by *RAD24* deletion but altered to only a minor extent by loss of Mec1 activity (**S3C and S3D Fig**). Importantly, however,

deletion of *RAD24* also skewed recombination events away from chromosome ends in the *msh2Δ* background (**S3E and S3F Fig**), which was itself similar to wild type (**S3A Fig**). Taken together, these findings suggest that enrichment in DSB [56] and recombination activity in EARs may independently depend on both the extension of prophase length and on Rad24 function. As was observed with the differences between wild-type and *ndt80*AR strains (**S4A and S4B Fig**), such differences between genotypes, however slight, were reproducible across all chromosomes (**S4C and S4D Fig**), building confidence in these trends and interpretations.

Globally increased recombination suggests that DSB formation may be increased above wild-type levels in *mec1-mn* mutants consistent with the role of Mec1 in DSB *trans*-inhibition [32]. Additionally, we previously concluded that Mec1 activation can promote DSB formation by inhibition of Ndt80 [16], indicating that Mec1 has antagonistic effects on recombination. Importantly, in the *mec1-mn ndt80AR* strain, delayed expression of Ndt80 causes cells to remain in prophase longer independently of Mec1 activation. Thus, the greater recombination that arises can be explained by de-repression of DSB interference in concert with extended time for DSBs to form. [16] Nevertheless, as when comparing any strains with severely affected spore viability, we cannot exclude that the observed increases are the consequence of selecting for a subpopulation of four-spore viable meioses, which in the *mec1-mn* background is associated with a hyper-recombination phenotype.

In comparison to the *mec1-mn* mutant, we observed differing changes in recombination frequency in *rad24Δ* depending on the strain background assayed. Firstly, in the single viable *rad24Δ sml1Δ* tetrad, CO and NCO rates were reduced to less than half of the wild-type values (**Fig 2B**). By contrast, the *rad24Δ ndt80AR* mutant displayed a ~1.3-fold increase in COs and NCOs compared to the *ndt80AR* control strain—a similar change to that observed in *mec1-mn ndt80AR* (**Fig 2C**). The similar recombination frequency in both *ndt80AR* DDR mutant strains contrasts with the very different frequencies observed between the DDR mutant strains with natural prophase length (**Fig 2B**). Although the inability to generate repeat datasets for *rad24Δ sml1Δ* hampers our statistical confidence, we tentatively conclude that the length of prophase has a greater positive impact on recombination frequency in *rad24Δ* than in *mec1-mn*.

Loss of *RAD24* function in the *msh2Δ* strain led to no increase in recombination frequency (**Fig 2D**). Instead, COs were reduced to ~75% of control levels ($P = 0.014$), and NCO frequencies were unchanged. These changes contrast strongly with both the impact of losing *MEC1* function in the *msh2Δ* background (above), and with the increased recombination observed in *rad24Δ ndt80AR* (**Fig 2C**). Such differing responses to loss of *MEC1* and *RAD24* in the *msh2Δ* background suggest that despite their known role in the DDR, Mec1 and Rad24 may function independently of one another under certain circumstances consistent with prior findings [37,38].

We reasoned that because *sml1Δ* does not compensate for the checkpoint or transcriptional functions of Mec1 [44], and because recombination frequencies were not significantly altered between wild type and *sml1Δ*, or between *rad24Δ sml1Δ msh2Δ* and *rad24Δ msh2Δ* (**S5 Fig**), the level of recombination in *rad24Δ sml1Δ* may be similar to that of *rad24Δ* single mutants (which were precluded from our analysis because we were unable to obtain any four spore viable tetrads). This interpretation agrees with the prior observation that CO numbers are reduced in *rad24Δ* strains [17,38].

## Increased occurrence of non-exchange chromosomes in DDR mutants

To aid the correct disjunction of chromosomes during meiosis I, it is important that each chromosome receives at least one CO per homolog pair—a process known as CO assurance [57,58]. To investigate whether this aspect of regulation is intact within Mec1 and Rad24

mutants, the frequency of non-exchange chromosomes was assessed (**Fig 3A and 3B** and **S2 Table**). We used a random simulation to evaluate the chance of observing given numbers of non-exchange chromosomes if there was no assurance (see Methods). Strikingly, the single four-spore viable *rad24Δ sml1Δ* tetrad contained six chromosomes with no detectable CO or NCO, significantly more than would be expected due to chance (*P* = 0.016, **Fig 3A and 3B**).

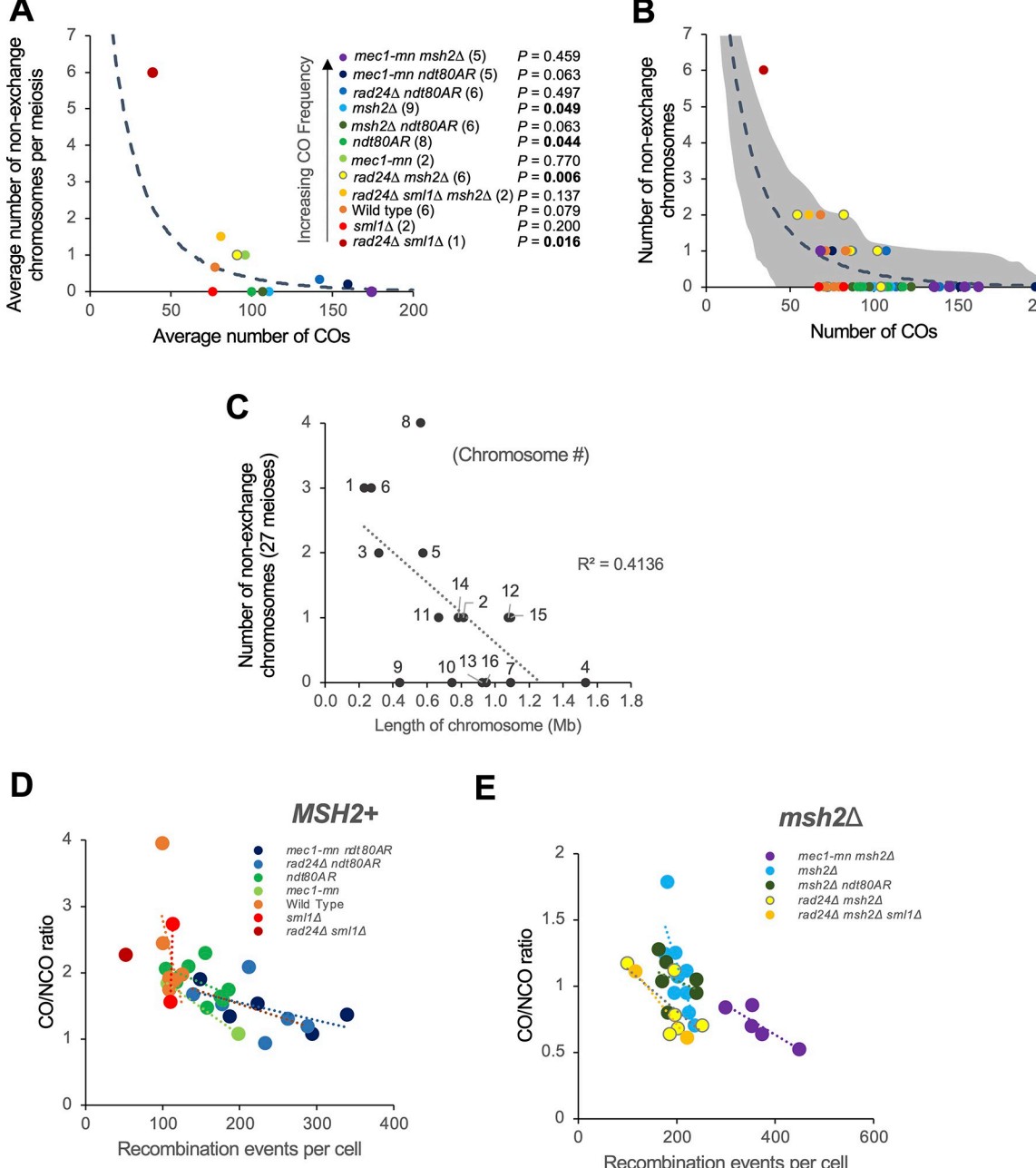

**Fig 3. Meiotic recombination event counts and spatial regulation in the absence of Mec1 or Rad24. A)** The average number of non-exchange chromosomes per octad/tetrad, plotted against the average number of crossovers. Dotted line represents random simulation (see Methods). **B)** The number of non-exchange chromosomes against the number of crossovers in each meiosis studied. The shaded area represents the area in which 90% of the simulations fell (smoothed). **C)** The relationship between the length of a chromosome and the number of times it was a non-exchange chromosome in 27 *mec1-mn* and *rad24Δ* mutant meiosis. **D,E)** The relationship between the recombination event count and the CO/NCO ratio in each cell. The plotted lines are linear models of all *MSH2* (**D**) or *msh2Δ* (**E**) strains.

Additionally, half of the twelve *rad24Δ ndt80AR* or *rad24Δ msh2Δ* tetrads lacked a CO on at least one chromosome, though these results are only significant in *rad24Δ msh2Δ* tetrads ($P = 0.497$ and $0.006$ respectively, **Fig 3B**). In contrast, neither *mec1-mn ndt80AR* ($P = 0.063$) nor *mec1-mn msh2Δ* ($P = 0.459$) backgrounds contained significantly more non-exchange chromosomes than expected by chance, suggesting that the mechanisms contributing to CO assurance are less dependent on Mec1 than on Rad24, which may help explain why *mec1-mn* mutants have a higher spore viability than *rad24Δ*.

The occurrence of non-exchange chromosomes tends to anti-correlate with CO counts (**Fig 3A and 3B**), and with chromosome size (**Fig 3C**), suggesting that a sufficiently high CO frequency will help to ensure that each chromosome receives a CO. However, even when accounting for the reduction in non-exchange chromosome frequency caused by high CO frequency, two genotypes showed significantly fewer than expected non-exchange chromosomes: *ndt80AR* and *msh2Δ* ($P = 0.044$ and $0.049$ respectively, **Fig 3A and 3B**). Such results suggest that crossover assurance is present in *ndt80AR* and *msh2Δ* backgrounds. Assurance may also exist in other genotypes, but our analysis did not detect a significant difference from random.

To examine CO homeostasis—the process that ensures a stable CO frequency despite variability in DSB number [59]—we plotted the ratio of CO:NCO formation against the total number of detectable recombination events (as a proxy for DSB number) for individual *MSH2*+ and *msh2Δ* meioses (**Fig 3D and 3E** respectively). CO:NCO ratios cannot be compared between *MSH2*+ and *msh2Δ* datasets due to the increased visibility of NCOs in a *msh2Δ* background. Evidence of CO homeostasis, shown as a negative trend on these plots, is detectable within all mapped *rad24Δ* and *mec1-mn* strains, suggesting that neither Rad24 nor Mec1 is essential for this process. Homeostasis within wild-type and *sml1Δ* strains could not be assessed by this method due to the low variability in total event frequency displayed within the assayed meioses (see **Fig 3D**). Importantly, although we detected supporting evidence for CO homeostasis in the absence of Rad24 and Mec1 activity, it should be noted that, when the total number of recombination events is low, CO outcomes (per event) may be favoured—not just due to homeostasis—but also because COs are likely to provide a selective advantage for accurate chromosome segregation, which is essential to produce viable haploid progeny. Finally, we note that any change in the observed CO:NCO ratio between strains could also arise due to alterations in heteroduplex tract lengths altering the visibility of NCO events. However, only relatively minor changes in NCO event lengths were detected in the absence of Mec1 or Rad24 relative to controls (see below), suggesting this is a minor consideration.

## Loss of *RAD24* and *MEC1* function increases the frequency of recombination events initiated by more than one DSB

Spo11 DSBs and the resulting CO events are spread across the genome more evenly than expected by chance due to DSB interference [31,32] and CO interference [30,35,60], respectively. Previously, Mec1 has been implicated in the process of *trans* DSB interference [32], whereas Rad24 is likely to have an indirect influence on CO distribution given its importance in helping to load the pro-class I CO factor, Zip3 [37].

In wild-type cells, a small fraction (~3%) of events are categorised as "complex" (**Fig 2**). This classification is used when the pattern of genetic change observed at a particular genomic location is hard to explain arising via a simple NCO or CO event involving only two chromatids (e.g. a double non-crossover, **S6A and S6B** etc). Notably, for both COs and NCOs, the fraction of such complex events was greater in the absence of Mec1 or Rad24 than in the relevant controls (**Fig 2B–2D**)—and thus may be explained by a loss of DSB and/or CO regulation.

To further investigate the role of Mec1 and Rad24 in this process, we analysed the complex fraction and assessed the frequency of recombination events that were compatible with initiation by more than one DSB, something that is expected to be infrequent in wild-type cells but frequent in cells lacking DSB interference [31]. Such 'multi-DSB' events are defined here as recombination events that can be explained by the formation of two or more separate Spo11-DSBs arising within 1.5 kb, and which are initiated on independent chromatids (**Figs 4A, S6** and **S7**).

Potentially, a second DNA break could be formed by something other than Spo11 during the repair of a first DSB. For example, the endonuclease Mlh1-Mlh3, which has CO resolution activity in meiosis, also exhibits DNA cleavage activity [61]. To reduce the potential impact of such factors upon the analysis, the classification as a multi-DSB event is conservative in nature. Specifically, double COs (dCOs) are only considered when they affect all four chromatids in a double reciprocal exchange. Although it is possible for a dCO to involve two or three chromatids, the resulting patterns cannot be unambiguously distinguished and so are not considered in this analysis. Additionally, a CO+NCO cluster is only included in the analysis when the NCO falls on a third chromatid. Double NCOs (dNCOs) are only included when occurring on two sister chromatids or on two homologues with perfect overlap (**Figs 4A** and **S6A-S6D**). Supporting our interpretation, multi-DSB events often arose at the same genomic location as population-average DSB hotspots as measured by Spo11-oligo mapping [62] (**S6A–S6D Fig**). Due to the stringency of this analysis, it is likely that the true number of multi-DSB events is greater than presented.

Multi-DSB events were relatively infrequent in wild-type cells (~5% of events), and were not significantly increased by *msh2Δ* or *ndt80AR*-mediated prophase extension, although combining both genetic manipulations together did yield the largest proportion (**Fig 4B**). By contrast, both *mec1-mn* and *rad24Δ* strains, in all genetic backgrounds assayed, displayed a significant (~3-fold) increase in the formation of multi-DSB events compared to the relevant control (**Fig 4C–4E**). Thus, DSB formation seems to be spatially regulated to a similar degree by both Mec1 and Rad24. Within multi-DSB events, the proportions of dCO, dNCO and CO+NCO events were not substantially altered between strains (**Fig 4B–4E**).

Events seemingly formed by multiple DSBs on different chromatids could potentially arise from multi-strand invasions, whereby the invading filament sequentially repairs using multiple chromatids [63]. To eliminate the possible influence of multi-strand invasions on our multi-DSB event calculations, we looked specifically at the formation of multi-DSB events containing a segregation pattern of 8:0 or 7:1 (**S6E–S6G Fig**). Because conversion on two sister chromatids at the same genetic locus is necessary to form these patterns, they are strong indicators of multi-DSBs arising on different chromatids (i.e "*trans*-DSB" events), and cannot be easily explained by multi-strand invasion. Additionally, 8:0 patterns that were not flanked by heteroduplex tracts were excluded to avoid the potential inadvertent inclusion of mitotic recombination events within this analysis. The occurrence of "*trans*-DSB" events was significantly increased in both *mec1-mn* and *rad24Δ* mutants compared to controls (**Fig 4F** and **S2 Table**). Note, however, because we observe an intermediate effect in *msh2Δ ndt80AR* relative to either single mutant, some of the increases observed in the *ndt80AR* background may be driven by Msh2-dependent nicking activity rather than by Spo11 alone.

Given that the total number of events were different in *mec1-mn* and *rad24Δ*, yet the proportion of event types were similar, differences in event types do not correlate with differences in event number. Furthermore, small changes in event numbers did correspond to large changes in complex event types, such as in *mec1-mn* mutants (**Fig 4C–4E**). These observations support and extend the known role of Mec1 in promoting *trans*-DSB interference [32], and suggest a similar role for Rad24. Taken together, these results highlight a failure of spatial regulation to control DSB formation within Mec1 and Rad24 mutants, marked by a loss of DSB

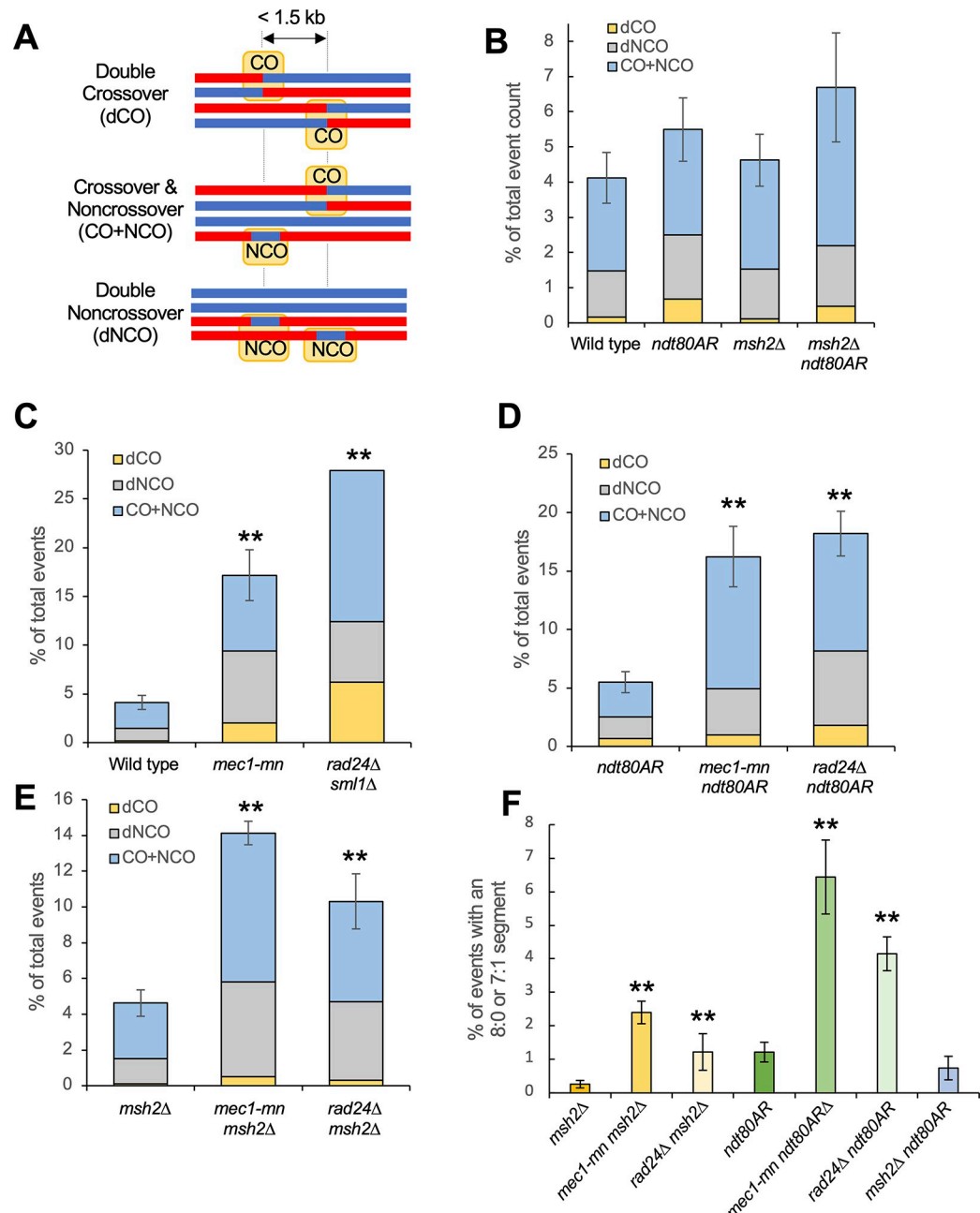

**Fig 4. Meiotic recombination frequency and spatial regulation in the absence of Mec1 or Rad24. A)** Simple depiction of the multiple-DSB event categories. **B-E)** Proportion of recombination events that are thought to result from multiple DSBs in **(B)** control strains, **(C-E)** *rad24Δ* and *mec1-mn* with the genetic background **(C)** WT or *sml1Δ*; **(D)** *ndt80AR*; **(E)** *msh2Δ*. The average proportion of multiple-DSB events per octad/tetrad is shown. Error bars are standard error of the mean. Differences in the proportion of multi-DSB events in each mutant and in the appropriate reference strain were tested by Fishers exact test (* = P<0.05, ** = P<0.01). **F)** The average number of directly overlapping events per octad/tetrad, defined as containing a segment of 8:0 or 7:1 segregation. All such events are also considered to be multi-DSB events of some kind. Differences were tested by Fishers exact test (* = P<0.05, ** = P<0.01). P values were corrected for multiple testing using the Benjamini-Hochberg method. Differences were not tested between *MSH2* and *msh2Δ* backgrounds because 7:1 segments are not generated in *MSH2* backgrounds.

interference in *trans* and an increased occurrence of nonexchange chromosomes, which impacts chromosome segregation and spore survival.

## Recombination tract lengths are increased in DDR checkpoint mutants

The number of genetic markers involved in a recombination event, and the total genomic distance that they span ('event length'), may be influenced by joint molecule migration (including, but not limited to dHJ branch migration), additional DNA strand breakage during repair [40], or by the distance of DSB resection, which averages ~1.5 kb in wild-type cells [64,65]. It is currently unclear to what extent each of these factors contribute to the final event length, or how DDR factors may contribute. Notably, hyper-resection at Spo11-DSBs has been observed in DDR mutants such as *rad24Δ*, *rad17Δ* and *mec1-mn* [10,16,18], suggesting that if DSB resection is a significant contributor to the overall amount of DNA involved in each recombination event, conversion tract lengths may be increased in *rad24Δ* and *mec1-mn* strains.

Because event lengths are affected by local marker density, estimates were made using the midpoint between the markers flanking each event (**Fig 5A**). To avoid the complexities inherent within overlapping multi-DSB events, only unambiguous single-DSB events were considered. To compare between samples, median recombination event lengths, along with upper and lower quartile values, were calculated for each genotype (**Figs 5B–5G** and **S8**).

Within control strains, *MSH2* deletion led to a significant decrease in event length, which was most pronounced for NCOs (**Fig 5B and 5C**), consistent with Msh2 acting to convert heteroduplex DNA arising within the nascent strand invasion intermediate [39]. Alternatively, mismatch rejection mediated by Msh2 at sites of invasion may increase event lengths by causing any subsequent strand invasion (and potential conversion) to occur at a more distally located site. Extending prophase length via *ndt80AR* moderately increased NCO event lengths of all quartiles (but not significantly), and CO event lengths of the upper quartile significantly ($P = 0.029$, Wilcox test), effects that were only observed in the *MSH2+* background (**Fig 5B and 5C**). Such effects may arise due to persistent MMR activity, and/or increased opportunity for branch migration of persistent recombination intermediates present in prophase-arrested cells [54].

Loss of Rad24 or Mec1 activity had complex effects on recombination event lengths (**Fig 5D–5G**). Whilst median lengths were broadly unaffected, longer recombination events were disproportionately extended—with the upper quartile event length values increased up to ~1.4-fold in DDR mutants compared to controls (**Fig 5D–5G**). These increases in event length may indicate that the hyper-resection evident in DDR mutants via physical analysis has an influence in the final pattern of recombination. It is also possible that DDR components influence other aspects of recombination, such as JM migration and/or stability, leading to longer tracts of genetic change in the haploid progeny. Intriguingly, NCO event lengths in the *msh2Δ* background are significantly shorter in both DDR mutants (**Fig 5G**). Thus it is possible that DDR components increase the stability of recombination intermediates—perhaps related to their known role in suppressing ectopic recombination [17–19] leading to shorter tracts of repair synthesis when mutated. Importantly, polymorphism density around NCO events was unchanged in the absence of Rad24 and Mec1 (~7 variants per kb), indicating that the reported changes in event length are not technical consequences of a redistribution of NCO events towards more highly polymorphic areas, which may have led to a more accurate estimate of event length.

## Detection of heteroduplex DNA in *rad24Δ* and *mec1-mn* mutant strains

Genome-wide analysis of meiotic recombination in *msh2Δ* octads permits the detection of not only hDNA, but also the determination of strand polarity, which enables a more detailed

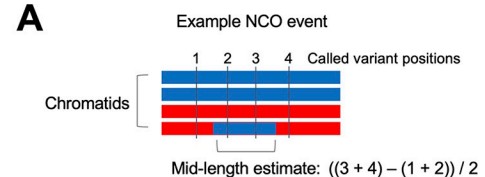

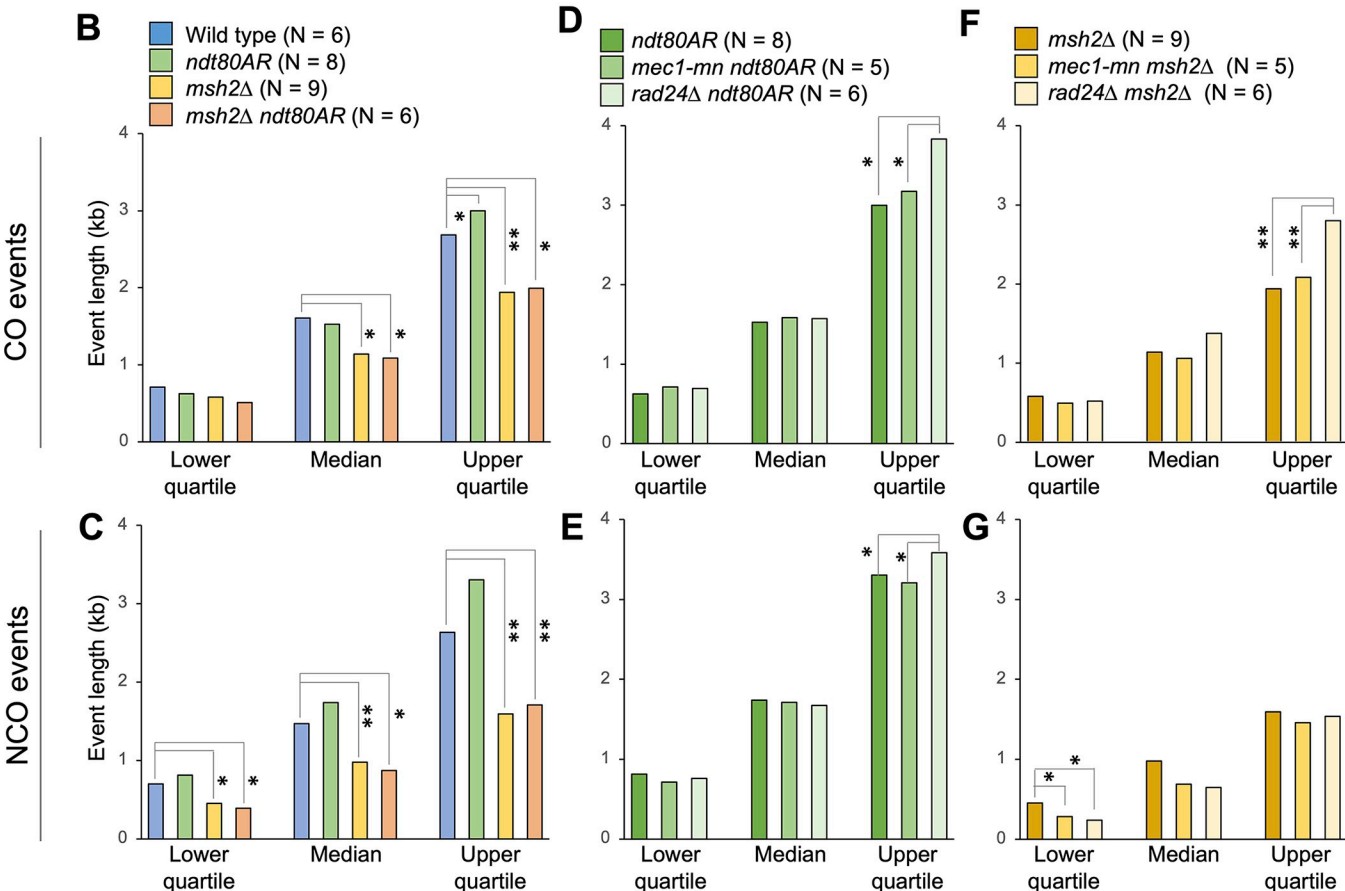

**Fig 5. Meiotic recombination event tract lengths in the absence of Mec1 or Rad24. A)** Simple depiction of event mid-length calculation. The mid-length estimate is defined as the distance between the midpoints of the inter-marker intervals surrounding the recombination event. The lengths of multi-DSB events are omitted. **B-G)** The median, upper and lower quartile values for the mid-lengths of the strand transfers associated with CO **(B,D,F)** and NCO **(C,E,G)** events in the indicated strains. Statistical differences between strains are indicated (Wilcoxon test, * = P<0.05, ** = P<0.01). P values were corrected for multiple testing using the Benjamini-Hochberg method.

inference of recombination mechanism [39,40]. Therefore, in order to investigate the role of DDR components in specific pathways of recombination, HR strand-transfer patterns were explored in more detail within *mec1-mn msh2Δ* and *rad24Δ msh2Δ* meiosis compared to controls, by computationally and manually sorting into categories as previously described [40]. Example events from each category are shown in **S9** and **S11 Figs**, and potential routes of formation are shown in **S10** and **S12 Figs**.

Direct analysis of heteroduplex patterns necessitates separation of mother/daughter cells following the first post-meiotic cell division of each member of a four-spore viable tetrad

(**Fig 6A**). Whilst formally possible, due to the low frequency of four-spore viable tetrads (**S4 Table**), it proved impractical to directly generate and analyse octads in *rad24Δ msh2Δ* and *mec1-mn msh2Δ* strains. Instead, heteroduplex information was recovered from *msh2Δ* tetrads, by converting markers displaying ~50% coverage of each genotype ("heteroduplex" calls) into two haplotypes each allocated a copy of each parental SNP (see Methods and [66]) to generate pseudo-octads (**Fig 6A**). Although this method allows the isolation of hDNA patches, information on strand polarity is not retrieved. To additionally recover strand polarity information, post-meiotic haploid cells were streaked onto rich media, and a single colony sequenced at high depth (**Fig 6A**, lower panel). Due to post-meiotic segregation, such clones will retain the complete haplotype information of one of the strands present in the mixed reads from the *msh2Δ* tetrad. By comparing this haplotype to the mixed heteroduplex reads, the haplotype of the other member of the pair can also be constructed [66]. Haplotype resequencing by this method was possible for a single sample of *msh2Δ rad24Δ* and *msh2Δ mec1-mn* meioses (**Fig 6B and 6C**) as described below.

Recombination events in nine control *msh2Δ* and six *msh2Δ ndt80AR* octads were categorised and compared with the reconstructed *mec1-mn msh2Δ* and *rad24Δ msh2Δ* octads. Notably, the proportions of NCO (**Fig 6B**) and CO (**Fig 6C**) events detected in each category were not significantly altered between *msh2Δ* and *msh2Δ ndt80AR* octads, suggesting that the prophase arrest does not impact the way in which DSBs are repaired. In the DDR mutants, the frequency of one-sided and two-sided NCOs were both significantly reduced, whereas full conversion NCOs were elevated (**Fig 6B**). One- and two-sided NCOs are thought to be produced by the SDSA pathway [40]. Full conversion patches (regions of 6:2 segregation) may be produced by single-stranded nicking during repair, or via the repair of gaps generated by adjacent Spo11 DSBs, suggesting that one or both processes are subject to regulation by Mec1 and Rad24, the latter of which is consistent with loss of DSB interference.

Interestingly, *rad24Δ msh2Δ* also displayed a small but significant increase in NCOs with changes on two non-sister chromatids (**Fig 6B**; 'Two Chr')—indicative of an increase in NCO formation by dHJ resolution—something that was not observed in *mec1-mn msh2Δ*. Note however, that these could also be indicative of two DSBs both undergoing NCO formation, although this is a more complex explanation. Because the proportion of these 'Two Chr' events are also more frequent in a *zip3Δ* [63], our results are consistent with Rad24 having a role in the loading of ZMM proteins at the sites of future CO events [37]. Specifically, the loss of ZMM loading in *rad24Δ* may mean that an increased proportion of joint molecules that would otherwise become class I COs are instead resolved by class II factors such as Mus81, which is thought to be unbiased in its production of CO and NCO events. Consistent with this view, the CO:NCO ratio was lower in *rad24Δ msh2Δ* than the *msh2Δ* control (see **Fig 3D**).

Loss of Mec1 and Rad24 function also had slightly different effects on hDNA patterns associated with COs. The *mec1-mn msh2Δ* octad displayed a significant reduction in the frequency of one-sided bidirectional COs and two-sided unidirectional COs than both the *msh2Δ* control and the *rad24Δ msh2Δ* octad (**Fig 6C**). Bidirectional CO patterns are thought to be produced by dHJ migration [40]—suggesting that dHJ migration is reduced in *mec1-mn msh2Δ*. Bidirectional CO patterns are also thought to be a signature of MutL-gamma (Mlh1-3)-dependent class I CO resolution [40], suggesting a defect in this pathway in *mec1-mn*. The only CO category altered in frequency in the *rad24Δ msh2Δ* octad was a decrease of COs with no associated strand-transfer pattern (**Fig 6C**). COs with no strand-transfer patterns are those occurring in regions of low marker density, and so have no associated hDNA information. Potentially, this could suggest a preference for COs in regions of higher marker density in *rad24Δ*.

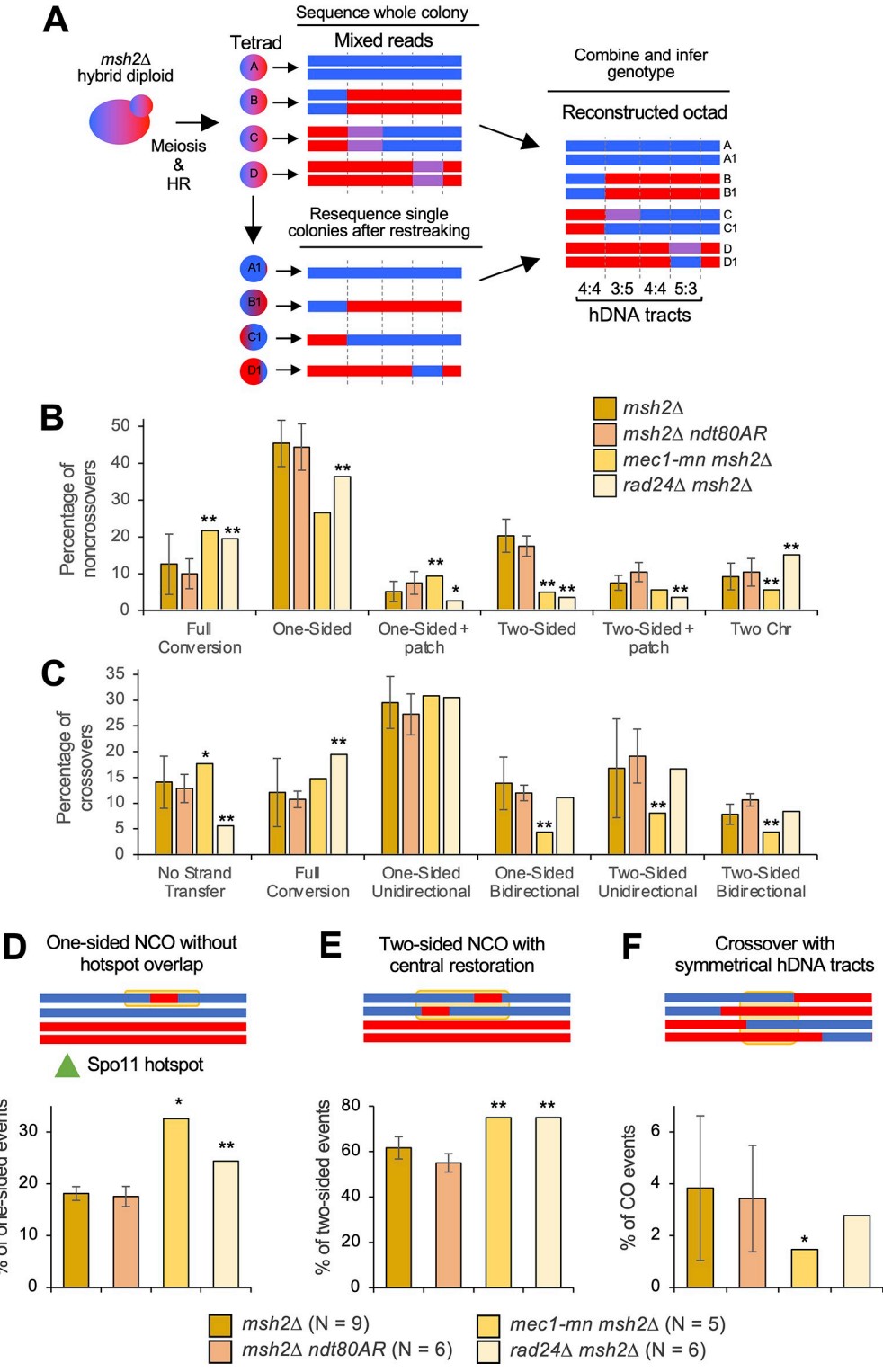

**Fig 6. Subcategorization of single-DSB events based on hDNA patterns. A)** Strategy for the retrieval of hDNA in low-viability backgrounds. *msh2Δ* strains are mated and sporulated as in **Fig 1A**, and an entire sectored colony is sequenced, containing mixed reads at positions of hDNA. Additionally, a restreaked single colony is sequenced; the genotype of this colony at each position containing mixed reads allows the genotype of the other strand to be inferred. **B,C)** Crossover **(B)** and noncrossover **(C)** categories in *msh2Δ*, *msh2Δ ndt80AR*, *rad24Δ msh2Δ* and *mec1-mn msh2Δ*. All error bars are standard error of the mean. Stars indicate when the proportion of multi-DSB events is significantly

different in the mutant and in the *msh2Δ* reference strain (one star, p < 0.05; two stars, p < 0.01, Fishers exact test). Only the *rad24Δ msh2Δ* and *mec1-mn msh2Δ* tetrads that were able to undergo detection of hDNA polarity were used (*mec1-mn msh2Δ* #3 and *rad24Δ msh2Δ* #6 in **S2 Table**). Crossover and noncrossover events were classified according to their strand transfer patterns, illustrated in **S9** and **S11 Figs**. The percentage of each category over the total number of crossovers or noncrossovers is shown. Complex event types (**S6 Fig**) contribute to the total number of COs and NCOs and thus percentages, but are not plotted again here. All categories are mutually exclusive. **D,E,F)** Proportion of events that appear to have involved template switching during repair. A simple example of each category is given, with the yellow box highlighting the region thought to have undergone sister chromatid invasion. **D)** One-sided NCOs that have occurred without overlapping a hotspot. **E)** Two-sided NCOs with a central patch of 4:4 restoration. **F)** Crossovers with symmetrical hDNA tracts. P values were corrected for multiple testing using the Benjamini-Hochberg method.

## The role of Mec1 and Rad24 in maintaining inter-homolog bias

Mec1, along with Tel1, promotes inter-homologue recombination in meiosis [67] via the phosphorylation of Hop1 [21]. We were thus interested to investigate whether recombination patterns in *rad24Δ* and *mec1-mn* mutants displayed evidence of reduced homologue bias. While it is not possible to directly observe inter-sister recombination using our assay, we reasoned that alterations in the frequency of template switching (initial repair with the sister chromatid, then switching to repair with the homologue) in DDR mutants may suggest changes in the strength of inter-homologue bias.

Events containing patches of putative template switching were categorised by the presence of unconverted markers in recombination segments expected to be converted during normal DSB repair (**Fig 6D–6F**, with potential recombination pathways shown in **S13 Fig**) [40]. Surprisingly, only one class was altered: the proportion of one-sided NCOs that occur away from a Spo11 hotspot, and this was only increased in *mec1-mn* (*P*<0.05) (**Fig 6D**). These observations suggest either that there is redundancy in Hop1 activation and the promotion of inter-homologue bias (Pch2 has been suggested to play such a role [68]) or that our analysis is not sensitive enough to detect major changes. Notably, *mec1-mn* strains display the greatest frequency of recombination events (up to ~360 in total in *mec1-mn msh2Δ*) of any strain analysed—indicating that, at least in those meioses analysed here, inter-homologue recombination is not impeded.

## The even spacing of crossovers is reduced in *MEC1* and *RAD24* mutants

An important aspect of spatial regulation in meiotic recombination is CO interference, which describes the observation that COs are spaced more evenly than would be expected by chance [35]. While the DDR component, Tel1, has been implicated in both DSB interference [31,32] and CO interference [69], roles for Mec1 and Rad24 are less clear, and may be distinct [37,38].

To examine the role of Mec1 and Rad24 in CO patterning, inter-CO distances (ICDs)—the distances (in bp) between successive COs along each chromosome—were calculated for *mec1-mn* and *rad24Δ* strains in both *msh2Δ* and *ndt80AR* backgrounds, arranged in order from smallest to largest and plotted as cumulative fraction of total CO events against ICD size (a cumulative distribution function, CDF, plot; **Fig 7A–7H**).

To evaluate the uniformity of CO patterns, experimental ICD distributions were compared to a random distribution. Gamma models were fitted to experimental ICD distributions, with higher shape parameters indicative of more even spacing, a feature that can arise via interference [70]. In all control strains, experimental ICDs displayed a steeper sigmoidal curve relative to the random simulation, indicative of more even spacing between CO events and therefore lower variance (**Fig 7A–7H**). Notably, loss of Msh2 activity led to a more pronounced skew away from random in this curve (**Fig 7A and 7B**), indicating more even spacing, and potentially indicative of stronger interference than in wild type. Furthermore, a similar effect of

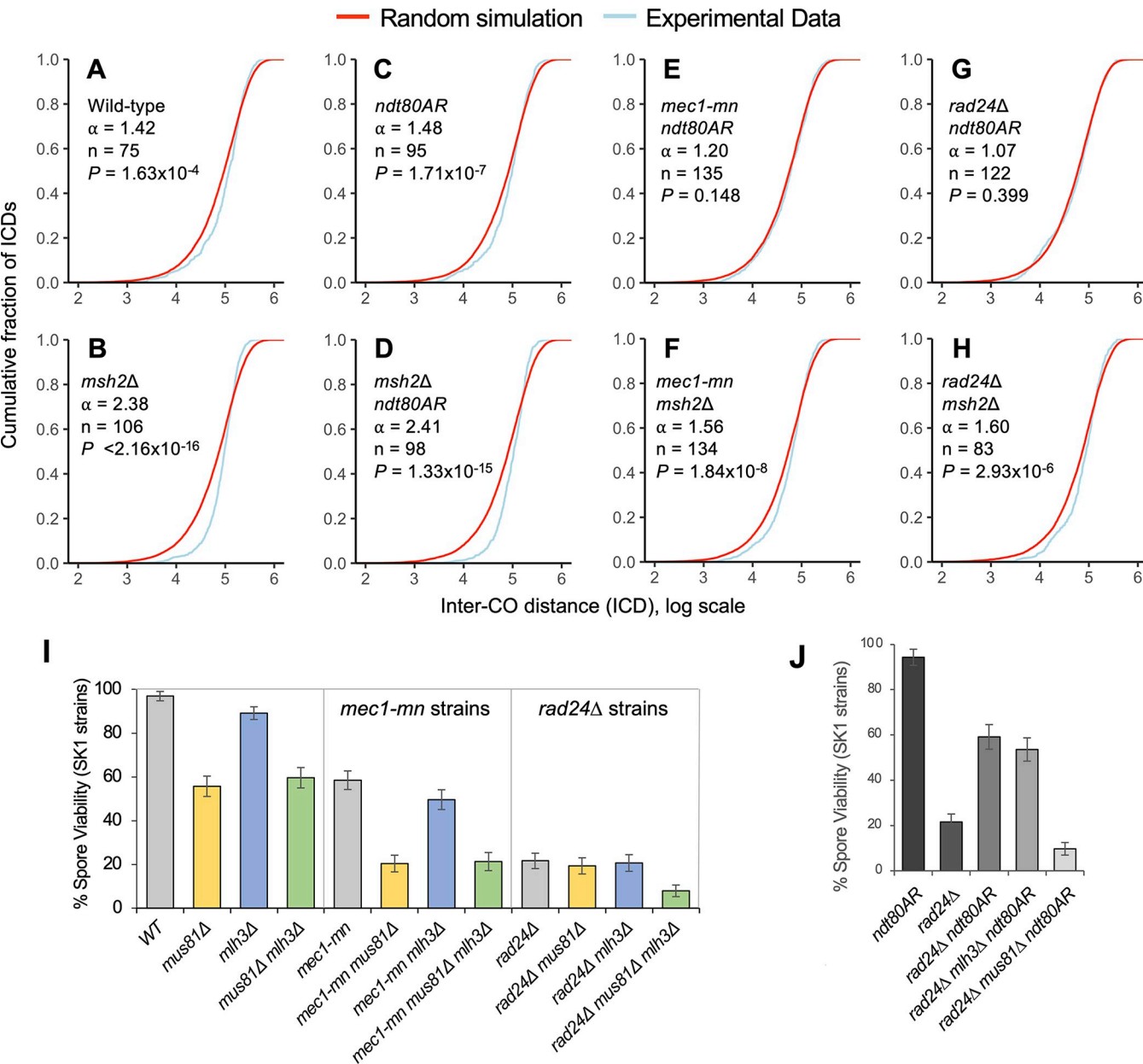

**Fig 7. Inter-CO distance in Mec1 and Rad24 mutants compared to control strains. A-H)** Inter-CO distances (ICDs) were calculated for the indicated strains, rank ordered, and plotted as cumulative fraction of the total CO count against inter-CO distance (ICD) on a log scale. For comparison, random datasets were generated from a gamma distribution with an α value of 1 with the same mean as the ICDs in each strain. α values given are from gamma distributions fitted to the experimental data from each strain using maximum likelihood estimation. Differences between experimental ICD distributions and those of the random simulation were tested using a Kolmogorov-Smirnov (KS) test. **I, J)** Impact of the loss of CO resolution factors Mlh3 (Class I COs) and Mus81 (Class II COs) on the spore viability of *rad24Δ* and *mec1-mn* SK1 strains. Spore viability comparison between **I)** WT, *rad24Δ* and *mec1-md* SK1 strains with or without Mlh3 or Mus81; **J)** *rad24Δ* SK1 strains with or without 8 hours of *ndt80AR* and Mus81/Mlh3. Error bars are 95% confidence limits.

*MSH2* deletion was observed in the *ndt80AR* background (**Fig 7C and 7D**). By contrast, prophase extension, mediated via the *ndt80AR* allele, had no effect on ICD distributions (**Fig 7A and 7C**), and there was also no change in the *msh2Δ* background (**Fig 7B and 7D**). Notably, shifts in CO distribution were independent of CO number (**Fig 7A–7D**).

By contrast, loss of either Mec1 or Rad24 activity led to a shift in ICD distribution towards that of the random simulation (**Fig 7C, 7E, 7G, 7B, 7F and 7H**), consistent with a reduction in

the global manifestation of CO interference. This shift was most pronounced in the *ndt80AR* background, where the distribution of ICDs was not significantly different from that of the random distribution (**Fig 7C**, **7E and 7G**). By contrast, the CO distributions, whilst less uniform than the *msh2Δ* control in both *msh2Δ mec1-mn* and *msh2Δ rad24Δ*, remained significantly skewed away from random, suggesting retention of some degree of CO interference (**Fig 7B**, **7F and 7H**).

## Differential dependence of Mus81 and Mlh3 recombination pathways in DDR checkpoint mutants *mec1-mn* and *rad24Δ*

ZMM-dependent (class I) COs are resolved by the Mlh1-Mlh3 complex [57,71,72] whereas class II CO formation requires the nuclease activity of Mus81/Mms4, Yen1 or Slx1-Slx4 [72–77]. Rad24 has also been proposed to have a specific role in ZMM loading [37]. To further investigate the recombination pathways that are functioning when *RAD24* and *MEC1* are absent, we compared spore viability in a variety of class I and class II CO mutants in combination with *rad24Δ* or *mec1-mn* (**Fig 7I**).

In the wild-type (control) background, loss of Mus81 activity led to a more severe reduction in spore viability than loss of Mlh3 activity (~56% viability and ~89% viability, respectively, versus ~97% in wild type ($P<0.001$ and $P = 0.0074$ respectively, *mus81Δ* against *mlh3Δ* $P<0.001$; **Fig 7I**). Interestingly, loss of both Mus81 and Mlh3 activity resulted in no additional spore viability reduction relative to the *mus81Δ* mutant ($P = 0.382$, **Fig 7I**)—an effect similar to that observed in *mlh3Δ mms4Δ* strains [78]. This suggests that loss of the class II CO pathway causes a greater reduction in viability than loss of the class I CO pathway. Moreover, it also suggests that loss of the class I CO pathway does not result in any further viability reduction in the absence of the class II CO pathway.

Compared to the wild-type control, *mec1-mn* strains exhibit a reduction in their baseline spore viability (~58%, $P<0.001$, **Fig 7I**). Similar proportional decreases in spore viability as in the wild-type background were observed for both *mus81Δ* and *mlh3Δ*, suggesting that the effects of losing Mus81 and Mlh3 pathways are retained in the absence of Mec1, and thus are independent of Mec1 activity (~20%, $P<0.001$ and ~50%, $P = 0.0044$, respectively, versus ~58% in the *mec1-mn* control; **Fig 7I**). Furthermore, as in the wild-type background, loss of both Mus81 and Mlh3 in the *mec1-mn* background resulted in no further reduction of spore viability relative to loss of Mus81 ($P = 0.798$, **Fig 7I**). Despite the overall similar trends observed in *mec1-mn*, there was, nonetheless, a slightly greater dependence on the Mus81 pathway in *mec1-mn* strains—with spore viability reducing ~3-fold rather than ~2-fold in the wild-type control background. This effect may indicate a greater reliance on the class II recombination machinery when Mec1 is absent.

To our surprise, and contrary to the above observations made in wild-type or *mec1-mn* strains, individual loss of either the Mus81 or Mlh3 pathways had little impact on spore viability in the *rad24Δ* background ($P = 0.544$ and $P = 0.814$ respectively, **Fig 7I**), whereas, combined perturbation of both recombination pathways resulted in a substantial decrease in spore viability (from ~20% to ~8%, $P<0.001$; **Fig 7I**). These results suggest that class I and class II factors may act redundantly in the absence of Rad24, allowing COs and/or other nascent recombination intermediates to be resolved by either pathway. Notably, though the baseline viability of *rad24Δ* strains was substantially lower than occurs upon Mec1 depletion (~22% versus ~58%, $P<0.001$; **Fig 7I**), the fact that we were able to detect the combined effect of losing both Mus81 and Mlh3 pathways, suggests that the overall low spore viability was not masking an effect, if such an effect existed, in the *mus81Δ rad24Δ* strain.

Importantly, the *rad24Δ* strains in which recombination patterns were analysed utilised *msh2Δ* and *ndt80AR* alleles to increase spore viability. Such an increase in spore viability likely arises, at least in part, by increasing the frequency of COs per meiosis (**Fig 2B–2D**), but which may then place an extra burden on the recombination machinery. To test this idea, we examined the impact on spore viability of removing either Mlh3 or Mus81 activity in the prophase-extended *rad24Δ ndt80AR* strain (**Fig 7J**). Consistent with the substantially random placement of COs in *rad24Δ ndt80AR* strains (as also in *mec1-mn* strains; **Fig 7E and 7G**), increases in spore viability in *rad24Δ ndt80AR* strains (mediated by the prophase extension) were largely independent of Mlh3 activity (~59% to ~54%), but very strongly dependent on the activity of Mus81 (~59% to ~10%).

## Comparisons between the loss of DDR components and Zip3 function

Loading of Zip3 to chromosomes is partially dependent upon Rad24, but not Mec1 [37] motivating us to compare the phenotypes of the *rad24Δ*, *mec1-mn* and *zip3Δ* mutants.

Direct comparisons made in the *msh2Δ* background revealed (like in *MSH2* strains [63]) that *ZIP3* deletion causes a very significant increase in the event length associated with meiotic COs (**Fig 8A**)—something that is, in relative terms, increased only slightly in *rad24Δ* and *mec1-mn* mutants (**Fig 8A**). A comparable difference in event length was not observed when considering NCOs (**Fig 8B**). Larger CO lengths within *zip3Δ* have been proposed to arise from an increase in Sgs1-dependent D-loop extension [63], suggesting that either the residual Zip3 that loads in *rad24Δ* is sufficient to inhibit Sgs1 but is insufficient to mediate class I CO formation, or that Rad24 is required for the spurious Sgs1 activity that arises when Zip3 is completely absent.

*zip3Δ msh2Δ* strains displayed a dramatic reduction in the ratio of COs to NCOs (**Fig 8C and 8D**), suggesting a loss of CO control. Notably, this effect was much stronger than in *zip3Δ* single mutant controls (**S2 Table** and [63]), suggesting that—at least in the absence of Zip3, when homologue engagement is defective and DSB formation persists [79]—Msh2 may indirectly promote CO formation via the suppression of excessive numbers of NCOs. Whilst *rad24Δ msh2Δ* and *rad24Δ msh2Δ sml1Δ* strains also showed disproportionately few COs, the effect was less severe than observed in *zip3Δ msh2Δ*, and, importantly, the paucity of COs in *rad24Δ* strains was largely restored by extending the length of meiotic prophase (**Fig 2C**). Taken together, these results suggest a much more significant defect in CO regulation upon loss of Zip3 function than upon loss of Rad24. We note, however, that to fully elucidate the mechanistic relationship between Zip3 and Rad24 in CO control will require the analysis of a *zip3Δ rad24Δ msh2Δ* triple mutant.

The overall frequency of recombination was also significantly elevated in *zip3Δ msh2Δ* compared to *msh2Δ* (~404 vs ~206 events; ~2-fold; **Fig 8C**), and similar to that observed in *mec1-mn msh2Δ* (~368 events)—albeit without the dramatic change in CO:NCO ratio. As mentioned above, ZMM-dependent homologue engagement is one of the triggers that suppresses DSB formation at later stages of prophase [79–81]. The similarity between the phenotypes of *zip3Δ* and *mec1-mn* in terms of increased total event frequency suggest that homologue engagement may elicit its negative regulatory effect via Mec1. In contrast, despite reduced levels of Zip3 loading within *rad24Δ* [37], no elevation in total event count was observed upon inactivation of this DDR factor (~188 events; **Fig 8C**).

Previously in *zip3Δ* mutants, an increase in NCO events likely to be formed from dHJ resolution was noted [63], proposed to result from unbiased cutting of dHJ intermediates in the absence of a pathway promoting CO formation. Here we recapture this result in *zip3Δ msh2Δ*. This strain displays not only a large increase in the absolute number of NCOs likely formed

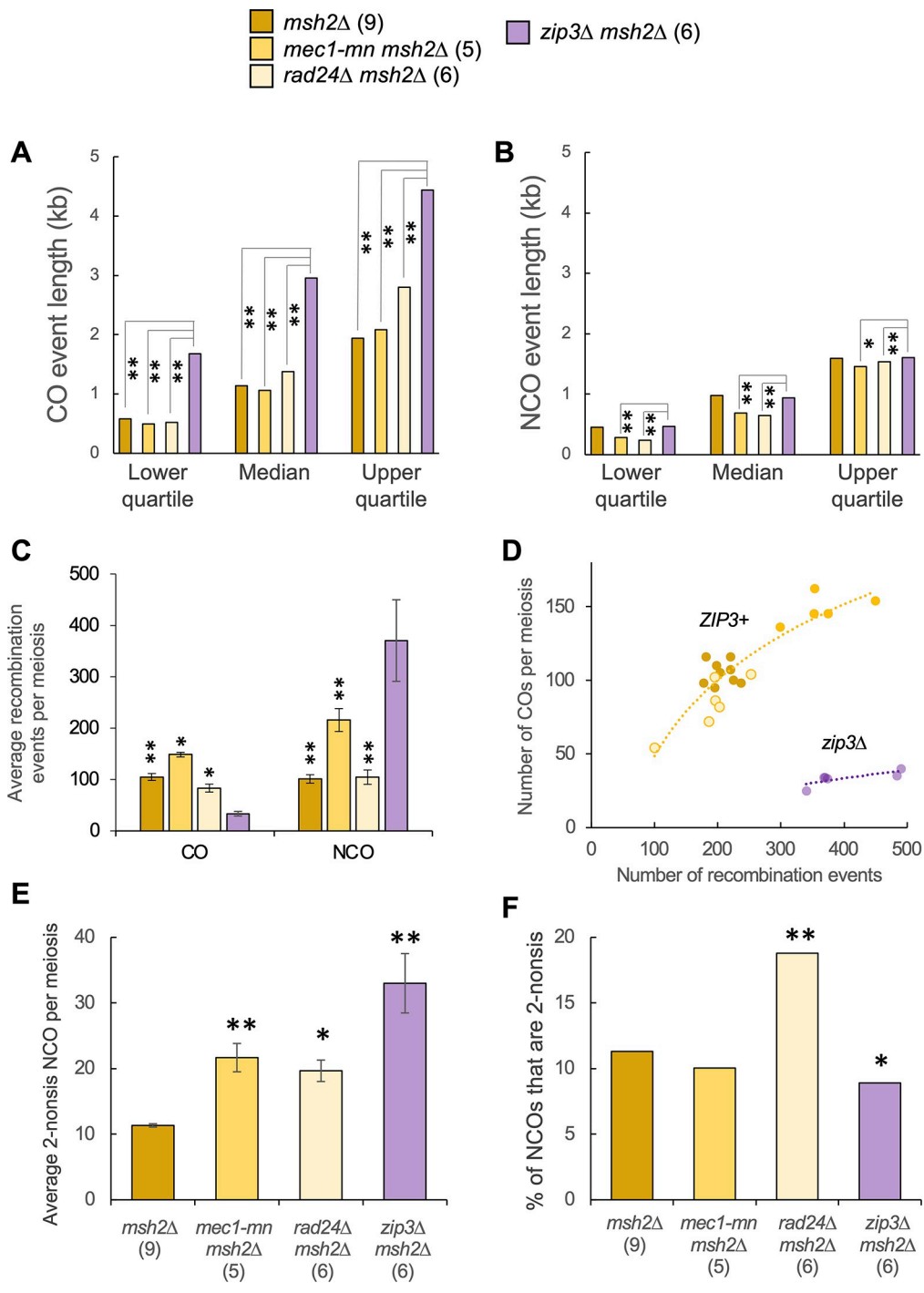

**Fig 8. Comparison of recombination event characteristics between DDR mutants and *zip3Δ*. A)** CO and **B)** NCO event lengths were ranked as a cumulative frequency distribution, and lower, median and upper quartile values presented as in **Fig 5**. Multi-DSB events were omitted. **C)** Recombination event counts. **D)** The relationship between recombination event counts and the number of CO events in individual meiosis. The lines are an exponential model for either the *zip3Δ msh2Δ* strain (purple) or all other (*ZIP3+*) strains (orange). **E)** The average number of NCO events that appear to have been formed by dHJ resolution (see **S6A** and **S7A Figs**). Differences to the reference strain were tested by T-test (*:P<0.05, **:P<0.01). **F)** The aggregate proportion of all NCO events that appear to have been formed by dHJ resolution (see **S6A** and **S7A Figs**). Differences to the reference strain were tested with Fisher's Exact Test (*: P<0.05, **:P<0.01). P values were corrected for multiple testing using the Benjamini-Hochberg method.

from dHJ resolution (**Fig 8E**), but the resulting frequency (33 per meiosis; **Fig 8E**) is very similar to the frequency of the remaining COs (33.5 per meiosis; **Fig 8C**). This similarity suggests a complete loss of CO bias during dHJ resolution upon loss of Zip3. However, overall, due to there being such large increases in total NCO number, only ~9% of NCOs in *zip3Δ msh2Δ* were formed from dHJ resolution, which was actually a significant reduction in proportion compared to *msh2Δ* (P<0.05) (**Fig 8F**). We speculate that this latter result is likely due to the substantial increase in DSB formation arising from failure to establish homolog engagement, producing many DSBs that are then repaired as NCOs via the SDSA pathway.

While there was also a significant increase in the absolute number of NCOs formed from dHJ resolution in *rad24Δ msh2Δ*, it was not as large as the increase in *zip3Δ* or even *mec1-mn* (**Fig 8E**). However, these NCOs formed a much larger proportion of total NCO counts (~19%) (**Fig 8F**). Taken together, these results indicate that in both *zip3Δ* and *rad24Δ* backgrounds, there is an increase in the formation of NCO events from dHJ resolution.

## Discussion

### Separable roles of Mec1 and Rad24 in regulating meiotic recombination

We report here the impact that loss of the DNA damage response proteins Mec1 and Rad24 has on many aspects of meiotic recombination (**Fig 9**). Rad24 is known to be important for activating Mec1 via its role in loading the 9-1-1 clamp in response to ssDNA produced by the resection of DSBs [2]. Thus, as examined here, it is not surprising that in some aspects of meiotic recombination the effects of *rad24Δ* are similar to those of *mec1-mn*. For example, both Mec1 and Rad24 limit total recombination frequency (**Fig 2B–2D**), and prevent the formation of multi-DSB events (**Fig 4**)—supporting the prevailing view that the Mec1 pathway mediates DSB interference in *trans* [32]. The more extreme phenotype of *mec1-mn* is likely explained by the fact that Rad24 is not the only activator of Mec1 [7,82–84].

However, in many other cases the phenotypes of *mec1-mn* and *rad24Δ* are less similar, indicating separable roles for the two proteins. For example, the *rad24Δ* mutant is more dependent upon extension of prophase length to maintain high levels of recombination (**Fig 2B–2D**) and spore viability (**Fig 1C**) than *mec1-mn*. In general, the patches of genetic change arising per recombination event (event length) were also larger upon *RAD24* deletion than in the *mec1-mn* strains (**Fig 5D–5G**), perhaps in part due to small differences in the extent of hyper-resection observed in these two strains [16]. Indeed, Rad24 may enact its role in regulating resection distance by loading the 9-1-1 clamp, which may act as a barrier or inhibitor to resection [85]. Alternatively or additionally, the effect on resection may be because Rad24 and Rad17 are involved in activating Mec1, which may then prevent resection by deactivation of Exo1 [86]. In vegetative cells, the 9-1-1 complex inhibits resection by promoting the recruitment of Rad9 near DSBs [87]; however, this may not be the case for meiotic DSBs, because Rad9 is not required for checkpoint activation in meiosis [10].

Event length differences may also arise from an alteration in recombination mechanism. In the meioses evaluated here, *rad24Δ* strains displayed some of the highest frequencies of non-exchange chromosomes (**Fig 3A**)—values that are likely to be underestimates of the true non-exchange chromosome frequency due to our selection for meioses generating four viable spores. Such errors are particularly pronounced in the *rad24Δ msh2Δ* and *rad24Δ sml1Δ* strains, perhaps because the lack of checkpoint activation leads to less time to initiate and/or repair recombination events. Indeed, this is supported by the relative lack of non-exchange chromosomes in the *rad24Δ ndt80AR* strain, in which prophase is extended, allowing more time for COs to form and thus reducing the number of non-exchange chromosomes. Supporting the concept that prophase length plays a positive role in securing accurate chromosome

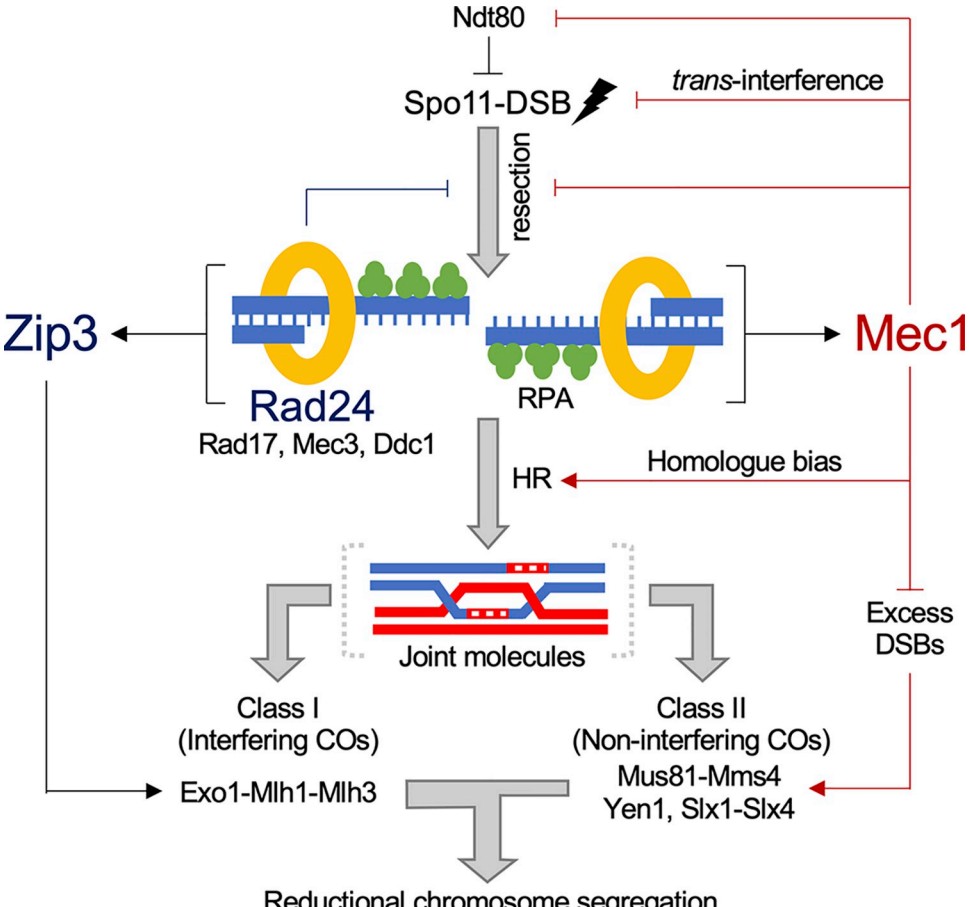

**Fig 9. Model of Mec1 and Rad24 activity influencing meiotic DSB formation and repair.** Proposed effects of Rad24 are indicated via dark blue lines, and those of Mec1 via dark red lines. Our results support and contribute to the view that Mec1 influences meiotic recombination in numerous ways: promotion of DSB formation via transient inactivation of Ndt80, suppression of DSB formation in *trans*, inhibition of hyper-resection, promoting homologue repair, and suppression of class II CO formation. Rad24 may support these roles by promoting Mec1 activation. In addition, Rad24 promotes Zip3-dependent class I CO formation, and/or the process of CO interference propagation, thereby driving the even distribution of COs along and between chromosomes via the process of CO interference. Together these pathways ensure accurate reductional chromosome segregation during meiosis.

segregation, we observed no non-exchange chromosomes in the *ndt80AR* single mutant strain, unlike in the wild-type control. Increases in non-exchange chromosomes may also arise from defects in the spatial patterning of DSBs, perhaps caused by defects in DSB interference in *mec1-mn* and *rad24Δ* strains. Moreover, across the dataset, non-exchange chromosomes often also lacked any detectable NCO, suggesting that DSB initiation defects (thereby affecting both COs and NCOs) manifest in a chromosome-autonomous manner. Given that Rad24 and Mec1 may affect DSB patterning via DSB interference, loss of such spatial regulation over when and where DSBs occur may underpin the result in achiasmate chromosomes.

Despite a similar checkpoint defect as *rad24Δ*, we infer that *mec1-mn* strains survive more readily than *rad24Δ* due to a derepression of DSB formation. As indicated above, we recognise that a potential limitation of whole genome recombination assays is the requirement for meioses generating four viable spores, leading to potential selection bias within analysed data. Nevertheless, it seems logical that such biases will tend towards less extreme, more wild type-like

phenotypes, suggesting that the characteristics we report here for *rad24Δ* and *mec1-mn* are likely to underestimate the true extent of the defects occurring during meiosis. Furthermore, because *mec1-mn* is a depletion (rather than a deletion), the phenotypes we report here may be less severe than *rad24Δ* for this reason.

Interestingly, the effects of *mec1-mn* described here are similar to those of *pch2* mutation [88], which has been found to cause global increases in DSBs and COs. *pch2* mutants also have a significantly lower incidence of non-exchange chromosomes, and CO interference appears weaker compared to wild type [88]. Similar phenotypes are described here for the *mec1-mn msh2Δ* and *mec1 ndt80AR* strains (**Fig 3A**). Whilst these observations may suggest a common pathway, synergistic effects of Mec1 and Pch2 on homologue bias indicate redundancy in Hop1 activation, and thus perhaps independent roles in regulating crossover formation.

## RAD24 and MEC1 deletion phenotypes are distinct from loss of Zip3

Rad24 is known to promote Mec1-independent loading of ZMM proteins at sites of future interfering COs [37]. The low spore viability of *rad24Δ* strains complicates direct study of the effects of *RAD24* deletion on CO distributions. However, since *ndt80AR* does not have a significant effect on CO distributions compared to wild type (even when CO numbers differ), we believe it reasonable to infer that CO distributions in the *rad24Δ* strain will not be significantly different from the *rad24Δ ndt80AR* strain described here, despite differences in CO numbers. Our results here suggest that Rad24 is one factor that helps to make CO distributions more evenly distributed than expected by chance, potentially by either increasing the proportion of interfering COs or by increasing the strength of CO interference.

In agreement with classic tetrad analysis, which showed both Rad24 and Mec1 to be important for CO interference [37,38], the CO distributions reported here are similarly affected by loss of either protein (**Fig 7C, 7E and 7F**). Thus, despite previously reported specific effects of Rad24 on ZMM loading [37] we cannot infer any differential effect of Rad24 and Mec1 in terms of effect on CO interference or CO type (proportions of class I versus class II) based on CO distributions alone. However, the reduction in the frequency of events with bidirectional CO patterns (which are a signature of MutL-gamma (Mlh1-3)-dependent class I CO resolution [40] in *mec1-mn*, do suggest a specific defect that is not present in *rad24Δ*.

Taking observations collectively, we speculate that the prior inference for a specific (Mec1-independent) role for Rad24 on ZMM loading may stem from the fact that global recombination frequencies (DSBs and CO outcomes) are lower in *rad24Δ* strains than in *mec1-mn*—potentially due to the loss of DSB trans interference in *mec1-mn* strains [32]. Thus, both Rad24 and Mec1 may have similar defects in class I CO formation (per DSB event), but that this defect is substantially rescued by a global increase in recombination (in absolute terms) in *mec1-mn* strains making it appear that ZMM loading is not substantially affected.

Supporting this view, loss of Zip3 function had a largely dissimilar phenotype to both *rad24Δ* and *mec1-mn*, suggesting that the primary defect in either DDR mutant is not loss of ZMM loading, but more likely a combined effect of this defect and loss of the meiotic prophase checkpoint. Moreover, no elevation in total event count is observed upon inactivation of *rad24Δ*, unlike in *zip3Δ* (**Fig 8C**)—suggesting that homologue engagement-dependent DSB suppression (which is perhaps mediated by Mec1) is still sufficiently active in the absence of Rad24.

In summary, the absence of either Rad24 or Mec1 causes COs to be spaced more randomly with greater variance in distances between them, and suggests that this is due to partial loss of Zip3 activity per recombination event in both strains.

## Rad24 and Mec1 affect CO distributions opposite and independently of Msh2

In contrast to the effects of *rad24Δ* and *mec1-mn*, *msh2Δ* alone makes CO distributions less random than in the wild type—something that is conserved in all *msh2Δ* background strains studied when compared to their controls (**Fig 7A–7H**). Notably, because both *rad24Δ msh2Δ* and *mec1-mn msh2Δ* have a more random CO distribution than *msh2Δ*, yet a less random CO distribution than that of either *rad24Δ* or *mec1-mn* (**Fig 7A–7H**), we conclude that the effects of Rad24 and Mec1 are independent of Msh2.

One potential explanation of these observations is that fewer interfering COs arise in the absence of Mec1 or Rad24, and that *MSH2* deletion partially rescues this defect. An alternative interpretation is that Rad24, Mec1 and Msh2 all affect the strength of interference, with Rad24 and Mec1 increasing interference strength, whereas Msh2 decreases interference strength. Finally, in budding yeast, distributions of DSBs—which are influenced by DSB interference [31]—may have a substantial impact on the resulting CO distributions. In these terms, the apparent reductions in the uniformity of CO distributions may arise from loss of DSB interference due to loss of Rad24 and Mec1.

In addition to altering CO distributions, *msh2* mutants also have greater numbers of COs compared to wild type and a 3-fold increase in NCOs [39,89–91]. Interestingly, the change observed in CO distributions in all *msh2Δ* strains, irrespective of Rad24 or Mec1 mutation, were independent of any change in the number of COs. Overall, our *MSH2* findings made in *S. cerevisiae* contrast observations in *Arabidopsis thaliana*, which instead suggest that Msh2 suppresses non-interfering crossovers, thereby reducing the strength of interference [90,91]. Such contrast between yeast and Arabidopsis could be due to differences in the regulation of meiosis and the vastly differing genomic architecture that exists between the two species (such as polymorphism density, chromosome length and number, length of time in prophase and number of COs; [89–91]).

## Meiotic roles of Rad24 and Mec1 orthologues in other organisms

Our findings indicate that Rad24 and Mec1 have both overlapping and distinct functions during *S. cerevisiae* meiosis. Current observations in other organisms suggest relatively conserved roles for ATR$^{Mec1}$ during meiosis (Introduction)—consistent with the coordination between meiotic DNA repair and meiotic chromosome segregation that is essential for fertility. However, there has been relatively little analysis of the roles of the checkpoint clamp and loader during meiosis outside of *S. cerevisiae*.

In *Schizosaccharomyces pombe*, Rad17 (the clamp loader) is required to delay meiotic onset in response to DNA damage, and is important to maintain normal levels of recombination in meiosis and spore viability [92]. Interestingly, *S. pombe* has no Zip3 orthologue, and all COs are non-interfering (class II), being resolved by Mus81 [93,94], suggesting that these defects cannot be mediated via a modulation of the CO interference pathway. In male mice, *Rad9a* mutants display abnormal testes with low sperm counts due to spermatocytes arresting in late zygotene or early pachytene [95]. These defects correlate with a deficiency in DSB repair leading to apoptosis, perhaps due to failure to initiate HR or because of a failure to activate the DDR checkpoint [95]. Similarly, *Hus1* mutant mice display severely reduced fertility and chromosomal abnormalities, but HUS1 is not required for meiotic functions of ATR in response to chromosome synapsis defects [96]. *Hus1* mutant female flies and worms are sterile [97,98], and *Drosophila* Hus1 is required for proper SC disassembly and efficient DSB processing by HR [99]. In plants, both RAD1 and HUS1 promote accurate DSB repair during meiosis, in particular suppressing nonhomologous end-joining [100] and ZMM-independent CO formation between heterologues [101].

Due to the interplay between various aspects of meiotic recombination regulation, it is somewhat difficult to ascertain which, if any, of these defects are distinct from a generalised abrogation of ATR function. By contrast, building on the work of Shinohara et al [37,38], our results highlight separable roles for Rad24 as a pro-CO factor, and for Mec1 as a regulator of recombination frequency—functions that are clearly specialised for the meiotic system. Moreover, given the essential role that COs play in chromosome segregation and fertility in most sexually reproducing organisms, it will be of great interest to clarify whether the DDR checkpoint clamp and/or loader play similar roles across other species as they do in *S. cerevisiae*.

## Materials and methods

### Yeast strains and culture methods

*Saccharomyces cerevisiae* strains used in this study are derivatives of SK1 [102], S288c [103], or BY4741, a derivative of S288c [104]. Hybrid diploid strains were generated by mating haploid SK1 with S288c or BY4741 parents. Strain genotypes are listed in **S3** and **S4 Tables**. Gene disruptions were generated by standard Lithium acetate transformation [105]. Gene disruptions of *rad24Δ::HphMX, msh2Δ::kanMX6, mus81Δ::kanMX6* and *zip3Δ::HphMX* were performed by PCR mediated gene replacement using pFA6a-*kanMX6* or pFA6-*hphMX* plasmids [106,107]. $P_{CLB2}$-*MEC1* strains ('*mec1-mn*') were created by replacing the natural *MEC1* promoter with the mitosis-specific *CLB2* promoter using pFA6a-*natMX4-PCLB2-3HA* plasmid as a template [16]. The $P_{GAL}$-*NDT80::TRP1* allele has the natural *NDT80* promoter replaced by the *GAL1-10* promoter, and strains include a *GAL4::ER* chimeric transactivator for β-oestradiol-induced expression [45]. The base *mlh3Δ* strain (*MATa ura3 lys2 ho::LYS2 arg4Δ* (*eco47III-hpa1*) *mlh3Δ::kanMX6*) was kindly provided by Michael Lichten.

### Meiotic timecourse with *NDT80* prophase arrest-release

Diploid strains were incubated at 30˚C on YPD plates for 2 days. For SK1 diploids, a single colony was inoculated into 4 ml YPD (1% yeast extract / 2% peptone / 2% glucose) and incubated at 30˚C (250 rpm) for 24 hours. For hybrid crosses, haploid parents are mated in 1 ml YPD for 8 hours, after which an additional 3 ml YPD was added, and the cells are grown for a further 16 hours. Cells were inoculated into 30 ml YPA (1% yeast extract / 2% peptone / 1% K-acetate) to a final density of OD600 = 0.2, and incubated at 30˚C (250 rpm) for 14 hours. Cells were collected by centrifugation, washed once in water, then resuspended in 30 ml pre-warmed sporulation media (2% potassium acetate, 5 μg/ml Adenine, 5 μg/ml Arginine, 5 μg/ml Histidine, 15 μg/ml Leucine, 5 μg/ml Tryptophan, 5 μg/ml Uracil), and incubated at 30˚C (250 rpm). As necessary, synchronized cultures were split after the required amount of time (e.g. 8 h) in 2% potassium acetate, and one fraction induced to sporulate by addition of beta-oestradiol to a final concentration of 2 mM. Cultures were then incubated to a total of 48 h at 30˚C prior to tetrad dissection. For all recombination mapping experiments using *ndt80AR* arrest-release, cultures were incubated for 8 hours in potassium acetate prior to oestradiol induction.

### Dissection of tetrads/octads to assay spore viability and for sequencing hybrids

To assay spore viability, 50 μl of sporulated cells in sporulation media were incubated with Zymolyase 100T (10 mM Sucrose, 0.7% Glucose, 1 mM HEPES pH 7.5, 1 mg Zymolyase 100T) at a final concentration of 4 μg/ml in a 150 mM sodium phosphate buffer at 37˚C for 15 min. 15 μl of digested cells was pipetted onto a YPD plate and allowed to dry before tetrad dissection. Dissected spores were incubated for 2 days at 30˚C on YPD and scored for percentage

viability per strain and viable spores per tetrad. To produce hybrid spores for genome sequencing, SK1xS288c and SK1xBY4741 haploid parents were mated for 8–14 hours on YPD plates, with the exception of *ndt80AR* strains, which were mated and grown in liquid YPD for 24 hours (see previous section). Haploids were mated freshly on each occasion and not propagated as diploids, in order to reduce the chance of mitotic recombination. Cells were washed and incubated in sporulation media at 30˚C with shaking, and tetrads were dissected after 72 hours. To generate octads, dissected spores were allowed to grow for 4–8 hours on YPD plates until they had completed the first post-meiotic division, after which mother and daughter cells were separated by microdissection, and allowed to grow for a further 48 hours (Fig 1A). Spore clones were subsequently grown for 16 hours in liquid YPD prior to DNA isolation. Only tetrads generating four viable spores, and octads generating eight viable progeny, were used for genotyping by Next Generation Sequencing. The haploid strains used for octad or tetrad sequencing are highlighted in **S3 Table**.

## Preparation of samples for sequencing

To determine the meiotic strand transfers that occurred in the absence of Msh2, we systematically generated and genotyped all members of each octad from S288C x SK1 hybrids. Genomic DNA was purified from YPD cultures of each haploid post-meiotic cell using standard phenol-chloroform extraction techniques. Genomic DNA was diluted to between 0.2–0.3 ng/µl, and concentration measured using the Qubit High Sensitivity dsDNA Assay. Genomic DNA was fragmented, indexed and amplified via the Nextera XT DNA library Prep Kit reference guide workflow according to the Best Practises recommended by Illumina. To check the fragment length distribution and concentration of the purified libraries, 1 µl of undiluted library was analysed on an Agilent Technology 2100 Bioanalyzer using a High Sensitivity DNA chip. Samples were pooled (16–24 per run), denatured by heating at 96˚C for 2 minutes, ice-chilled and mixed with denatured PhiX control DNA at 1% final concentration. Sequencing was performed in-house using Illumina MiSeq instruments using paired end 2x300 bp or 2x75 bp cartridges.

Tetrad analysis was carried out on *ndt80AR* and WT strains, while octad analysis was performed for *msh2Δ* strains. Notably, spore viability in *mec1-mn msh2Δ* and *rad24Δ msh2Δ* backgrounds was still too low for post-meiotic separation to be efficient. Instead, heteroduplex DNA information in *msh2Δ* tetrads was partially retained by harvesting and processing entire colonies as a single sample and categorising variants present in a roughly 50:50 ratio (**Fig 6A**). This approach allowed the reconstruction of a 'mocktad' containing almost all recombination event information that would be present in true octad analysis, with the notable exception of DNA strand orientation. In addition, one *rad24Δ msh2Δ* and one *mec1-mn msh2Δ* tetrad were each restreaked into single colonies and a single colony derived from each of the spores resequenced (**Fig 6A**). This strategy enabled the full recovery of DNA strand orientation information for these samples (Michael Lichten, pers. comm.).

## Alignment of paired-end reads, detection of SNPs and indels and creation of SK1 genome

Spores were sequenced to an average sequencing depth of 45x. Paired read files are aligned using bowtie2 [108] to both the S288c reference genome (version R64-2-1_20150113) and a custom SK1 reference genome described below. Also included in the reference file are sequences from the yeast mitochondrial (GenBank: KP263414.1 [109]) and 2µ plasmid (GenBank: V01323.1 [110]). To optimize the alignment for long reads and tolerance of mismatches expected in the hybrid genome, the bowtie2 alignment is performed with the following

settings: -X 1000—local—mp 5,1 -D 20 -R 3 -N 1 -L 20 -i S, 1, 0.50. To create a custom SK1 genome, SNP and indel polymorphisms were detected in the S288c alignments using the GenomeAnalysisToolkit (GATK) function HaplotypeCaller [111]. The in-house program 'VariantCaller' then combined the GATK calls from 120 samples, in order to calculate the call frequency, total read depth and averaged variant read-depth:total read-depth ratios for each variant. Variants were filtered for a call-frequency between 45–55% of spores, a total read-depth spanning the site of >250 and where 95% or more of the reads at that site contained the variant. Variants located in repeated regions, long terminal repeats, retrotransposons and telomeres were discarded. The final filtered list yielded 64,581 SNPs and 3946 indels. Consistent with polymorphisms being randomly distributed across the *S. cerevisiae* genome, inter-marker distances approximated to an exponential distribution with a mean and median of 169 bp and 81 bp (93.12% of inter-variant distances are <500bp), respectively. The final variant list was then used to automatically produce a custom 'SK1mod' genome, using the S288c genome as a backbone and converting any SNP or indel positions into the newly detected SK1 equivalent. Because this method utilises the S288c reference as a scaffold, called SK1 variants are limited to SNPs and short indels, and therefore lacks any larger structural rearrangements or large insertions/deletions that may exist between the strains.

## Genotype calling in tetrads and octads

Sequence alignment (SAM) files were converted into a sorted binary (BAM) file using the Samtools view command [112], for downstream processing. The PySamStats module 'variation' (https://github.com/alimanfoo/pysamstats) was used to produce a list of the number of reads containing an A, C, T, G, insertion or deletion for each genomic position specified in the S288c or SK1 reference, for each spore clone. These whole genome coverage files were filtered using the SNP/indel list derived above and further processed using custom scripts. Genotypes were assigned according to the following rules:

i. All positions must have a read depth of at least 5. ii) A SNP is called as the SK1 variant if 70% or more of reads at that position match the variant, or as the S288c reference if 90% of reads match the reference. iii) If the variant and reference reads are above 90% of all reads *and* within 70% of each other, the position is called as 'heteroduplex'. iv) Insertions and deletions are called as having the variant genotype if 30% or more of reads at that position match the variant. This low threshold is used because the alignment of indel sequences is biased towards the reference sequence, which means that they are unlikely to be erroneously called as matching the variant genotype. For an indel to be called as the reference genotype, at least 95% of reads must match the reference sequence, and there must be fewer than two reads matching the variant call. This is because even if there is an indel, there are usually reference reads recorded as well due to the difficulty in alignment, so the presence of more than one variant read reduces confidence in the ability to correctly call the position as reference. For this reason, indel positions also cannot be called confidently as heteroduplex. For octads, each of the eight progeny were processed. For tetrads, the four spores were processed and the data for each spore duplicated in order to appear as an octad in order to simplify downstream processing. In octads and MMR-proficient tetrads, SNPs called as heteroduplex were discarded. However, in MMR-deficient tetrads, heteroduplex calls are converted as described below. To improve the fidelity of this method, SNP positions found to contain mixed reads across multiple (>2) samples were eliminated—presumed to be residual errors in the variant table. For each position with a heteroduplex call, the original 'mother' is converted to SK1 and the duplicated 'daughter' to S288c. This is an arbitrary choice since it is impossible to know which should have which call, assuming there is no bias in the directionality of the repair. The exceptions are

*rad24Δ msh2Δ* #6 and *mec-mn msh2Δ* #3 in **S2 Table**, in which a single member of the mother-daughter pair were resequenced in addition to the mixed reads (see above). This latter strategy enables the haplotype of the other member of the mother-daughter pair to be inferred. In all cases, the genotype calls are converted into a binary signal, either 1 for S288c or 0 for SK1.

### Event calling

Event calling utilized a custom Python script as described below [39,40]. Using the binarized input, chromosomes were split into segments with the same segregation pattern [40]. Also recorded is the segment type, which will be 1:7, 2:6, 2:6*, 3:5, 4:4, 4:4*, 5:3, 6:2, 6:2* or 7:1 as described [39]. Recombination events were called as being a set of segments located between two 4:4 segments longer than 1.5 kb [40]. A 4:4 segment corresponds to a Mendelian segregation profile, 5:3 and 3:5 segments to half-conversion tracts, 6:2 and 2:6 segments to full-conversion tracts, etc. For general analysis 8:0 segments with no associated flanking hDNA were assumed to be premeiotic in origin and excluded. Each recombination event can contain between 0–2 COs or NCOs. Additionally, we encounter events which cannot be classified because they occur at the end of the chromosome and never return to a 4:4 pattern. These are given type 'U'. Fig 2A–2D indicate the total number of COs and NCOs present in the events (counting two COs or NCOs when necessary). The events are also classified by the number of chromatids involved: one chromatid, two chromatids, either sister or non-sister, and three or four chromatids. All recombination event images are provided (**supporting data file** available at https://figshare.com/s/89100901b905324fc50f and https://doi.org/10.25377/sussex.27188226.v1). Alongside the recombination event strand patterns, the Spo11 profile and hotspot strength values [113], Rec114/Mer2/Mei4 (RMM) profile [114], Rec8 peaks [115], and transcribed regions [116] are plotted. Chromatid strand orientation in octads was reconstructed as described [40] and where it could be determined, is displayed in event images (**supporting data file** available at https://figshare.com/s/89100901b905324fc50f and https://doi.org/10.25377/sussex.27188226.v1). Events were further subclassified according to strand transfer patterns present (**S6** and **S8** **Figs**) as described [40]). Patterns include restoration patches, full-conversion patches and bidirectional conversions.

### Event tract length determination

The event mid-length, defined as the distance between the mid-points of the first and last markers to be converted at each end of the event, was used as an estimate of the true event tract length (**Fig 5A**). Multi-DSB events are excluded from the median calculation due to the difficulty in determining how much of the recombination tract length is attributable to each component event.

### Quality control regarding mitotic events

One *mec1-mn ndt80AR* tetrad (#6) contained a number of events that appeared to be mitotic. The tetrad in question contained three large (>100kb) regions of 8–0 segregation on chromosomes 4, 7 and 10 (**S14A Fig**), as well as several smaller 8–0 regions (TCMN6 event images in **supporting data file** available at https://figshare.com/s/89100901b905324fc50f and https://doi.org/10.25377/sussex.27188226.v1). These are considered likely to be premeiotic conversions due to the lack of accompanying hDNA, and the tetrad was thus excluded from most analyses except for the examination of event tract lengths in the rest of the genome. One *mec1-mn msh2Δ* tetrad (#1) also had an apparent premeiotic event, although of a different nature. The presence of heteroduplex reads along the entire length of chromosome 8 in two of the four

spores suggests that these two spores contain two copies of chromosome 8, one from each parent, suggesting that a duplication of the S288c copy of the whole chromosome had occurred just prior to entering meiosis (**S14B Fig**). This tetrad was also excluded from other analyses except for event tract lengths on the unaffected chromosomes.

As 8:0 segregation patterns may arise from rare mitotic recombination events occurring prior to meiosis, 8:0 segments were discarded from analyses if they had no adjacent hDNA, or appeared in the same locus across multiple meioses—suggesting mis-annotation within one of the parental genomic references. Moreover, to limit the possibility of mitotic recombination events contaminating meiotic patterns, haploids were mated freshly for each experiment, rather than propagating the strains as diploids, and mating before sporulation was limited to <14 hours.

### Determining hotspot overlap of one-sided noncrossover events

The maximum possible start and stop coordinates for each one-sided NCO event were used and compared to a list of Spo11-DSB hotspots [113]. If a hotspot intersected with the maximum possible event region, the event was considered to have hotspot overlap.

### Simulation of non-crossover chromosomes

To assess how the frequency of non-exchange chromosomes differs from random chance, a random simulation was conducted. Each of the 16 chromosomes of *S. cerevisiae* was assigned a probability equal to the ratio between the length of the chromosome and the total length of the *S. cerevisiae* genome. Crossovers were then randomly allocated to each of these chromosomes until the specified number of crossovers had been allocated (1–200 crossovers). From each simulation, the number of chromosomes in which no CO arose (non-exchange chromosomes) was recorded. The simulation was repeated 10,000 times for every number of COs from 1 to 200 (2,000,000 simulations total). For each meiosis, we determined the chance of observing an equal or greater/lesser number of non-exchange chromosomes in the random simulation, given the number of total crossovers observed in that meiosis. i.e. a meiosis with 60 observed crossovers was compared to the results of the random simulation with 60 crossovers. Each meiosis was then ranked by the number of non-exchange chromosomes. The median probability was then taken and raised by the power of the number of meioses to evaluate the difference between the genetic background and random chance, which minimizes the influence of outlier meioses. Chances of observing more or equal numbers of non-exchange chromosomes and chances of observing fewer or equal numbers of non-exchange chromosomes were considered independently. Data from this simulation is shown in **Fig 3A** and **3B**.

### Inter-crossover distance distributions and gamma modelling

In order to model CO distributions, inter-crossover distances (ICDs) were fit with a gamma distribution function—a continuous probability distribution characterised by two independent parameters: (i) $(\gamma)\alpha$ (shape factor) (ii) $(\gamma)\beta$ (scale factor). Both parameters can be derived from the mean and standard deviation of values in the distribution. A value of $(\gamma)\alpha = 1$ indicates an exponential distribution i.e. randomness. $(\gamma)\alpha$ values of >1 indicate skews towards greater uniformity, which is a feature of interfering distributions. Greater values of $\alpha$ (and lower standard deviation) suggest a stronger effect of interference [70].

For comparison to our data, random gamma distributions ($\alpha = 1$) were generated using the same mean as the mean ICD of the data. These random distributions were sampled (N = 10,000 cells) for comparison with our data. All simulations, by design, precisely matched

the event count experimentally observed for any given genotype/cell to allow for direct comparison with a random distribution.

## Statistical analyses

*Fisher's exact test* was performed with the R environment (http://www.Rproject.org/) using the R function fisher.test(),with the two-sided option. *Wilcoxon's test* was performed with the R environment using the R function wilcox.test(), with the two-sided option and a continuity correction. *Student's T-test* was performed with the R environment using the R function t.test() and the two-sided option. *Kolmogrov Smirnov (KS) test* was performed with the R environment using the R function ks.test(). The Benjamini-Hochberg method was used to correct for multiple testing, using the R function p.adjust from the stats package. Random gamma distributions were generated in the R environment using the R function rgamma() with shape = 1.

## Supporting information

**S1 Fig. Bioinformatics pipeline for whole-genome mapping of meiotic HR events.** The method is based on that described previously [39,40]. Notable differences are that mismatches are tolerated during the initial alignment to preserve reads containing variants from both parents; the alignment is carried out against both parental genomes, and genotype calls reconciled; and the variant table includes both SNPs and indels. Additionally, we introduce the technique of sequencing *msh2Δ* tetrads and using heteroduplex calls to reconstruct an ersatz octad.
(PDF)

**S2 Fig. Methods used to increase *rad24Δ* and *mec1-mn* spore viability.** All error bars are standard error of the mean. **A)** Rescue of *rad24Δ* spore viability by *sml1Δ*. The spore viability of both SK1 and SK1xS288c hybrid strains with a deletion of *RAD24* is improved by the additional deletion of *SML1*. **B)** The effect of increasing prophase length on the spore viability of hybrid *mec1-mn* and *rad24Δ* yeast. Hybrid SK1xS288c strains are arrested in prophase and released after 4–10 hours using an inducible *NDT80* system 'ndt80AR'. Both Mec1 and Rad24 mutants display an improvement in spore viability when prophase is extended (compare with non-arrested viabilities shown in **Fig 1C**), and the level of improvement correlates with the length of the arrest.
(PDF)

**S3 Fig. Prophase arrest skews recombination event formation towards telomeric regions.** The cumulative fraction of all recombination events plotted against the distance from the nearest telomere **(A,C,E)** or centromere **(B,D,F)**, for the indicated strains.
(PDF)

**S4 Fig. Prophase arrest skews recombination event formation towards telomeric regions.** The cumulative fraction of all recombination events plotted against the distance from the nearest telomere, stratified by chromosome, for the indicated strains. Each chromosome is represented by a single line, coloured by strain.
(PDF)

**S5 Fig. Meiotic recombination frequency is unchanged by *sml1Δ*.** The mean counts of CO and NCO events per tetrad are shown (including both single and multi-DSB events). All error bars are standard error of the mean. Event count differences between *sml1Δ* and *SML1* versions of strains were tested by two-tailed T-test, and found to be insignificant.
(PDF)

**S6 Fig. Examples of events matching multi-DSB categories.** All example images are from *msh2Δ* octads. **(A-G)** A representative set of multi-DSB events taken from *msh2Δ* octads. Horizontal lines represent the eight strands of DNA present during recombination, while vertical lines are SNP/indel locations, with the S288C and SK1 alleles coloured red and blue, respectively. The orange/purple background highlights the CO/NCO event region respectively; events containing both a CO and an NCO are also coloured orange. The bottom part of panels **A-D** shows the counts of immunoprecipitated Spo11-FLAG oligos for each position, using S288C coordinates [62] smoothed using a 101bp hann window. **A)** Double noncrossover affecting two sister chromatids. **B)** Double non-crossover affecting two non-sister chromatids; this is determined by the occurrence of a small region of double-exchange, which could potentially be compatible with a very close double CO. **C)** Double CO event; this is determined by the double reciprocal exchange. **D)** CO and NCO event; this is determined by the occurrence of the NCO on a separate chromatid to the CO event. **E)** CO and NCO event containing a region of 7:1 segregation. **F)** Double noncrossover affecting two sister chromatids, and containing a region of 8:0 segregation. **G)** Double NCO containing a region of 7:1 segregation.
(PDF)

**S7 Fig. Recombination intermediates for events matching multi-DSB categories. A)** Double NCO on two sister chromatids formed by two independent SDSA reactions (**S6A Fig**). **B)** Double NCO on two non-sister chromatids. A second event needs to be invoked to account for the 6:2 conversion on the right: either another DSB + SDSA or a nick repair during resolution (**S6B Fig**). **C)** Double CO. Determined by the double reciprocal exchange (**S6C Fig**). For CO + NCO (**S6D Fig**), typical DSBR with CO resolution as in **S9D Fig** plus an additional SDSA with nick translation. **D)** CO + NCO with 7:1 segment (**S6E Fig**). **E)** Double NCO with 8:0 segment. Two independent SDSA reactions initiated on two sister chromatids but with processing of both ends yielding a gap. The short hDNA tracts are not detected due to low SNP density. (**S6F Fig**). Double NCO with a 7:1 segment (**S6G Fig**) caused by multiple DSBs: The first causing a two-sided NCO as in **S10C Fig** on chr 1, the second causing SDSA with one nick translation on chromosome 4, the third causing SDSA with template switching on chromosome 2.
(PDF)

**S8 Fig. Meiotic recombination event tract lengths in the absence of Mec1 or Rad24. A)** Event mid-length calculation is defined as the distance between the midpoints of the intermarker intervals surrounding the recombination event. Multi-DSB events are omitted from these analyses. **B-G)** Distribution of mid-lengths of the strand transfers associated with CO **(B,D,F)** and NCO **(C,E,G)** events in the indicated strains. Red horizontal lines indicate the values of the Upper Quartile (U), Median (M) and Lower Quartile (L) summarised in **Fig 5**.
(PDF)

**S9 Fig. Examples of events matching NCO categories.** All example images are from *msh2Δ* octads. **(A-F)** A representative set of NCO events taken from *msh2Δ* octads. Horizontal lines represent the eight strands of DNA present during recombination, while vertical lines are SNP/indel locations, with the S288C and SK1 alleles coloured red and blue, respectively. The purple background highlights the NCO event region. Categories are named as in [40]. The term 'Two-Sided' refers occurrence of strand transfer patterns on both sides of the putative DSB location. A lack of hDNA on both sides of the event ('One-Sided') may be caused by an absence of markers, or may be due to a peculiarity of the repair process e.g. template switching. **A)** Noncrossover with a single full-conversion tract. **B)** One-sided noncrossover with a half-conversion tract. **C)** One-sided noncrossover with a half-conversion tract and an internal

patch of full conversion. **D)** Two-sided noncrossover with two half-conversion tracts (*trans* hDNA) affecting the same chromatid. **E)** Two-sided noncrossover with two half-conversion tracts affecting the same chromatid and separated by a restoration tract. **F)** Noncrossover that affects two non-sister chromatids.
(PDF)

**S10 Fig. Recombination intermediates of NCO events. A)** Full conversion. The short hDNA tracts near the edges of the gap escape detection due to SNP density (**S9A Fig**). **B)** Synthesis dependent strand annealing (SDSA). Results in a one-sided NCO (**S9B Fig**) or one-sided + patch if there is a nick translation within the converted strand of an hDNA tract (**S9C Fig**). **C)** Double SDSA. Results in a two-sided NCO (**S9D Fig**) or two-sided + patch if there is a nick translation within the converted strand of an hDNA tract (**S9E Fig**). Can also arise from dHJ dissolution and nicked dHJ disassembly. **D)** DSB repair with NCO resolution resulting in an NCO that affects two non-sister chromatids (**S9F Fig**).
(PDF)

**S11 Fig. Examples of events matching CO categories.** All example images are from *msh2Δ* octads. **(A-F)** A representative set of CO events taken from *msh2Δ* octads. Horizontal lines represent the eight strands of DNA present during recombination, while vertical lines are SNP/indel locations, with the S288C and SK1 alleles coloured red and blue, respectively. The orange background highlights the CO event region. Categories are named as in [40]. The term 'Two-' or 'One-Sided' refers to the occurrence of strand transfer patterns on one or both sides of the putative DSB location. The term 'Directionality' refers to whether markers from only one parent are converted, or both. Bidirectionality may be caused by multiple DSBs/nicking, or junction migration. 'Sym 4:4' refers to symmetrical hDNA, which may originate from HJ branch migration. **A)** Crossover with no detectable strand transfer. **B)** Crossover with a single full-conversion tract. **C)** One-sided, unidirectional crossover with a single half-conversion tract. **D)** One-sided, bidirectional crossover. **E)** Two-sided, unidirectional crossover with trans hDNA tracts on the two recombining chromatids. **F)** Two-sided, bidirectional crossover with trans hDNA tracts on one chromatid only.
(PDF)

**S12 Fig. Recombination intermediates of CO events. A)** DSBR CO resolution. If SNP density is not high enough, hDNA tracts are missed (no strand transfer, **S11A Fig**). If SNP density is high enough, hDNA tracts are detected (two-sided unidirectional, **S11E Fig**). **B)** DSBR CO resolution resulting in full conversion (**S11B Fig**). Both ends are processed. Short hDNA tracts not picked by the SNP density. **C)** DSBR CO resolution. Asymmetry in the positioning of the two HJ with respect to the initiating DSB event yeilding asymmetric hDNA tract (one long one short) (**S11C Fig**). The short hDNA tract is not detected due to low SNP density. There are many complex combinations of events that can give rise to one-sided bidirectional transfer (**S11D Fig**), so this had been omitted. **D)** Two-sided bidirectional event caused by two double-strand breaks, resulting in two-sided bidirectional transfer (**S11F Fig**).
(PDF)

**S13 Fig. Formation of single-DSB events. A**) DSB and gap widening, common steps giving rise to single-DSB events. **B)** differential formation of single-DSB events. **i)** One-sided NCO without hotspot overlap (**Fig 6D**). **ii)** Two-sided NCO with central restoration (**Fig 6E**). **iii)** Crossover with symmetrical hDNA tracts (**Fig 6F**).
(PDF)

**S14 Fig. Abnormal mitotic recombination events in *mec1-mn* strains.** Whole chromosome recombination patterns were mapped in hybrid strains by detection of SK1 (blue) or S288c

(red) markers. The region of interest is plotted as eight horizontal lines corresponding to the eight strands of DNA present in the original hybrid diploid, with vertical tick marks indicating called variant positions. **A)** partial chromosome conversions on Chr 4, 7 and 10 in a $P_{CLB2}$-*MEC1* +10 h tetrad (TCMN6); **B)** A duplication of S288c Chr 8 in a $P_{CLB2}$-*MEC1 msh2Δ* tetrad (TCMM4). Raw reads indicate the frequency of reads containing SK1 or S288c type polymorphisms detected at each position; these are translated into binary calls, which can only be SK1 or S288c.
(PDF)

**S1 Table. *S. cerevisiae* spore viability measurements.** Relevant homozygous diploid genotype (column 1), haploid strain numbers used to create each diploid strain (column 2), or SK1 diploid strain where applicable, genetic background (column 3), and number of tetrads dissected (column 4) are indicated. Mean spore viability (column 5) was scored as the percentage of dissected spores that show visible growth after 48 hours incubation at 30˚C. To estimate how well the measurement represents the viability of the population, 95% confidence intervals were calculated (column 6). Viability pattern (columns 7–11) refers to the percentage of dissected tetrads that displayed 4, 3, 2, 1 or 0 viable spores. The colour scale used in column 5 indicates the mean spore viability reported for each sample: (Green > Yellow > Red; indicating High > Medium > Low). For all recombination mapping experiments using *ndt80AR* arrest-release, cultures were sporulated for 8 hours prior to oestradiol induction.
(PDF)

**S2 Table. Key characteristics of meiotic recombination in sequenced meioses.** For each individual meiosis, values are given for the number of single CO and NCO events, the identity of chromosomes without a CO or NCO event, the number of multi-DSB events (either dCO, dNCO or CO + NCO), and the number of individual events containing an 8:0 or 7:1 segment (not an exclusive category; all such events are also considered to be multi-DSB events of some kind). An increase in the formation of multi-DSB events suggests a loss of CO interference and/or *cis/trans* DSB interference. The occurrence of chromosomes without COs suggests a loss of CO assurance; COs are not spread out evenly between chromosomes. Many of these chromosomes are also missing NCOs, meaning they could not have compensated for the lack of CO events. The occurrence of 7:1 or 8:0 segregation patterns within an event suggests a loss of *trans* DSB interference. Example images for multi-DSB and 8:1/7:1 segment events are shown in **S6 Fig**. 7:1 segments cannot be called in *MSH2* strains (NA) due to lack of heteroduplex strand information. * *mec1-mn msh2Δ 1* and *mec1-mn ndt80AR* 6 were not used for any analysis except event lengths, due to large mitotic duplication events (see **Methods** and **S14 Fig**). *rad24Δ msh2Δ* #6 and *mec-mn msh2Δ* #3 underwent haplotype resequencing (**Methods**) to reproduce a full octad (see **Fig 5A**).
(PDF)

**S3 Table. *S. cerevisiae* strains used in this study for tetrad/octad sequencing.** All strains displayed are haploid, and were mated immediately prior to sporulation and tetrad dissection.
(PDF)

**S4 Table. *S. cerevisiae* strains used in this study to measure spore viability.** All strains displayed are diploid.
(PDF)

**S1 Data. Figure data.** Underlying data used to make main and supplementary figures.
(XLSX)

**S2 Data. Master event table.** Unprocessed data used to create **S1 Data**.
(XLSX)

**S3 Data. TCMN6 annotated Images.** Example event images for one of the meioses studied.
(PDF)

## Acknowledgments

We thank Michael Lichten for strains and methodological suggestions, and Scott Keeney for useful discussions regarding DSB homeostasis.

## Author Contributions

**Conceptualization:** Margaret R. Crawford, Bertrand Llorente, Matthew J. Neale.

**Data curation:** Margaret R. Crawford, Jon A. Harper, Matthew J. Neale.

**Formal analysis:** Margaret R. Crawford, Jon A. Harper, Tim J. Cooper, Matthew J. Neale.

**Funding acquisition:** Matthew J. Neale.

**Investigation:** Margaret R. Crawford.

**Methodology:** Margaret R. Crawford, Jon A. Harper, Marie-Claude Marsolier-Kergoat, Bertrand Llorente, Matthew J. Neale.

**Project administration:** Matthew J. Neale.

**Resources:** Bertrand Llorente, Matthew J. Neale.

**Software:** Margaret R. Crawford, Jon A. Harper, Tim J. Cooper, Marie-Claude Marsolier-Kergoat, Bertrand Llorente.

**Supervision:** Matthew J. Neale.

**Validation:** Margaret R. Crawford, Jon A. Harper, Matthew J. Neale.

**Visualization:** Margaret R. Crawford, Jon A. Harper, Matthew J. Neale.

**Writing – original draft:** Margaret R. Crawford, Jon A. Harper, Bertrand Llorente, Matthew J. Neale.

**Writing – review & editing:** Margaret R. Crawford, Jon A. Harper, Bertrand Llorente, Matthew J. Neale.

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
