## [Decision Letter · Decision Letter 0]

26 Aug 2024

Dear Matt,

Thank you very much for submitting your Research Article entitled 'Separable roles of the DNA damage response kinase Mec1ATR and its activator Rad24RAD17 during meiotic recombination' to PLOS Genetics.

The manuscript was fully evaluated at the editorial level and by independent peer reviewers. The reviewers appreciated the attention to an important topic but identified some concerns that we ask you address in a revised manuscript. In addition, I have some comments, which are listed below just before the reviews. In addition, reviewer 3 expressed concerns, in the comments to editor, that some of the conclusions stated in the Results section would be more appropriate for the Discussion section, although they did not specifically state which conclusions should be moved--so you should decide for yourself.

We therefore ask you to modify the manuscript according to the review recommendations. Your revisions should address the specific points made by each reviewer.

To resubmit, log into your Editorial Manager account and select the option 'Revise Submission' in the 'Submissions Needing Revision' folder.

Yours sincerely,

Michael Lichten, Ph.D.

Academic Editor

PLOS Genetics

Eva Stukenbrock

Section Editor

PLOS Genetics

Reviewer's Responses to Questions

**Comments to the Authors:**

AE's comments:

L26        "and has less effect on spore viability"

L213-216    Almost all of the skew is in the first 40kb from the telomere, whereas my impression is that EARs are substantially larger. Is the increase uniform across chromosomes, or is it possible that a few chromosomes are dominating the signal? Perhaps this is a subtelomere thing, not an EAR thing?

L1375 and ff    please proof the reference list for capitalization and italicization uniformity

Table S1    The spore viability distributions don’t add up to 100%. What do these numbers represent? This table would benefit from a more extensive legend.

Figures general    In Table S1, viability is reported for several arrest periods for ndt80AR. Which of this arrest periods were used for the mapping experiments? This should be stated in figure legends or in a prominent place in the results section.

Statistics    In addition to stating what error bars represent, please indicate in each figure legend how many replicates were performed when error bars indicate standard error. If < 3 replicates were performed, then error bars should denote range.

Reviewer #1: The edits to the manuscript have largely addressed the issues I raised in my initial review. The problems with ascertainment bias obviously persist but the authors were very careful in highlighting the associated caveats throughout the manuscript.

Reading through the revised manuscript, I only noticed a couple of things:

1. One item that needs to be addressed is missing statistical analyses. In particular the results shown in Figure 2 are discussed at length regarding increases and decreases but, at least in some of the panels (e.g. 2C), it is not clear how significant these differences are. In addition, in figures where statistical analyses were performed (e.g. 4, 5…), p value thresholds should be corrected for multiple hypothesis testing. If such correction was already performed, please indicate it in the methods and the figure legends.

2. The other thing I was wondering about is the increased cell-to-cell variation, particularly for COs for some of the mutants. The authors propose that this heterogeneity may be related to asynchronous meiotic induction (line 200-202). I wonder whether the variation may instead be related to temporal differences in homolog engagement/chromosome synapsis among chromosomes. This seems testable because one would expect that excess COs will be non-randomly distributed among chromosomes, with some chromosomes receiving substantially more COs than others, but the affected chromosomes would differ from cell to cell. The author note later in the text that E0 chromosomes also lack detectable NCOs, which would seem consistent with such chromosome-by-chromosome effects. If detected, these effects would also impact the conclusions about the randomization of inter-crossover distances in Figure 7.

Reviewer #2: The authors have thoroughly addressed the concerns by reviewers and associate editor. I have only some minor points that the authors may want to fix:

L160: normal prophase I length

Fig. 3A: Could you please move the legend into Fig 3A, so it is closer to the dots? Direct genotype labeling of the dots with increased E0 chromosomes might be an alternative. As it stands now, it is really challenging for the reader to find the right label in the list, even there are only 4-5 genotypes that result in increased E0 events.

L407: isn’t dHJ migration a subset of JM migration? Maybe: “JM migration, including, but not limited to dHJ branch migration”

L513 & L526 & L724: Direct evidence for the role of Mec1 in homolog bias was provided by Joshi et al (2015; PMID: 25661491), and should be cited here. Synergistic effects of Mec1 and Pch2 on homolog bias further address the issue of redundancy in Hop1 activation (see L724).

Fig. 7A-E: Please bring the legend (Simulation/Experimental) further down into the Figure and increase the font size. Reading on the screen, it’s easy for the legend to get cut off, and this reader thought they have to figure out for themselves what red and blue curves stand for.

L576: Brown et al (2013; PMID: 23316435) should be cited which also reported the surprising lack of mlh3D effects on spore viability in an mms4 mutant background.

L605: Please rewrite this rather convoluted sentence.

Fig. 8E,F: Please refer the reader to the appropriate (supplemental) diagram for explanation of the y-axes: “Average 2-nonsis NCO per meiosis” and “% of NCOs that are 2-nonsis”. “2-nonsis” is not explained anywhere in the main manuscript, so only the true aficionados will be able to understand this figure.

L654: Please improve readability of this sentence: “due to increase…due to failure”

Reviewer #3: To clarify each function of the 9-1-1 clamp loader Rad24 and Mec1ATR, which are involved in DNA damage response, in controlling the meiotic recombination processes, the authors used a novel approach, deep sequencing of tetrads or octads after PMS in spores derived from S288C/SK1 hybrid diploids and in a mismatch repair-deficient background. Although many of the results merely obtained confirm those already reported, the novel methodology described above is interesting, even if there is room for further improvement, and is expected to provide valuable insights that will provide opportunities for future applications in various research fields. On the other hand, there have already been many reports showing that mismatch repair deficiency has a significant impact on the biochemical reaction of homologous recombination. Furthermore, this paper does not fully explain what occurs during meiotic recombination in the msh2Δ strain. Therefore, previously, I pointed out that the results obtained by meiotic recombination in the genetic background of msh2Δ do not necessarily coincide with what actually occurs in wild-type strains, and that the experimental results should be interpreted with caution. In this revision, this point is mentioned more appropriately in the text, and it has been improved. However, since the significant increase in spore viability of mec1-mn and rad24Δ strains in the msh2Δ background was accompanied by large alternations in the CO/NCO balance, we should avoid making detailed mention of the CO control analysis (Fig. 3), which used these CO/NCO ratios as parameters. Above all, this paper shows a specific contribution of MSH2 to CO interference (L549-551), which may lead to a recombination reaction that differs from that in the wild-type strain. Furthermore, the data from the rad24Δ sml1Δ strain, which represents a sample in which the six chromosomes are miraculously precisely distributed during meiosis I without COs, raises serious concerns regarding the generalizability of the observations. As a result, the paper, especially the first half, requires assumptions to interpret the data, making it difficult to generalize and understand what the data truly shows.

On the other hand, this method has successfully analyzed unusual meiotic recombination derived from multiple DSBs and DSB resection tract length in the DDR mutants. I agree the importance of making the data obtained public, but since making definitive claims of novel ideas based on data whose reproducibility has not been confirmed can lead to confusion. Authors should separate clear conclusions based on the experimental results from uncertain interpretations based on assumptions and consider keeping discussion of the latter more modest and simpler, or excluding discussion of data that is difficult to generalize to know the function of DDR factors in meiotic recombination.

Major points:

L240-242: Rather than mec1 deficiency resulting in greater recombination diversity, could this result be due to the possibility that tetrads selected for viability in mec1-mn cells survive by various means?

L245-248: Although the statistical significance of the difference is unclear, at least in the mec1-nm ndt80AR cells, recombination near the telomere appears to be higher than in the wild type. Also, in rad24Δ, recombination appears to be higher than in the wild type up to 40 kb from the telomeres (Fig. S3C). Please check the data. Therefore, the conclusion is an overstatement, I think.

L253-254: I don't understand the logic behind this conclusion. In the ndt80AR strain, Ndt80 is ectopically expressed at a later time than usual, so Mec1-dependent inhibition of Ndt80 does not occur at the time of Ndt80 recovery as in the wild-type. (Are there any data showing that inhibition occurs in the ndt80AR?) Rather, since the recombination frequency is even higher in the ndt80AR mec1-mn than in the ndt80AR, a function of Mec1 that is not dependent on Ndt80 should be considered.

L318-320: Isn't this explanation only valid under the assumption that the amount of DSBs in each strain is constant? The definition of CO homeostasis is that the frequency of COs per bivalent remains constant despite fluctuations in the amount of DSBs. Since NCO varies with the number of DSBs [1], please provide evidence that the amount of CO per NCO can be used as a criterion for CO homeostasis even when the amount of DSBs is not constant among the strains. At least, as you have discussed, there is a possibility that bias is being applied at the time of selecting surviving spores in rad24Δ. In other words, it is thought that complex events are occurring in each mutant background, so unless the analysis is performed in combination with the spo11 hypomorph mutations, it is difficult to simply consider the function of DDR factors in CO homeostasis with this method.

L549-551: More even spacing than in the wild type? What specific situation are you thinking of?

L633-634: Although its relationship with ZIP3 is still unclear, RAD24 has been reported to function upstream of ZIP1 in meiotic recombination. The rad24 mutation may bypass ZIP3 function upstream [2]. Analysis of the zip3 rad24 msh2 triple mutant is required to reach this conclusion.

Minor points:

L233: Please briefly explain the rationale for using the sml1Δ mutation.

Ref:

1. Martini E, Diaz RL, Hunter N, Keeney S. Crossover homeostasis in yeast meiosis. Cell. 2006;126(2):285-95. Epub 2006/07/29. doi: S0092-8674(06)00859-2 [pii]

10.1016/j.cell.2006.05.044. PubMed PMID: 16873061; PubMed Central PMCID: PMC1949389.

2. Shinohara M, Bishop DK, Shinohara A. Distinct Functions in Regulation of Meiotic Crossovers for DNA Damage Response Clamp Loader Rad24(Rad17) and Mec1(ATR) Kinase. Genetics. 2019;213(4):1255-69. Epub 2019/10/11. doi: 10.1534/genetics.119.302427. PubMed PMID: 31597673; PubMed Central PMCID: PMCPMC6893372.

**Have all data underlying the figures and results presented in the manuscript been provided?**

Reviewer #1: Yes

Reviewer #2: Yes

Reviewer #3: Yes

PLOS authors have the option to publish the peer review history of their article (what does this mean?). If published, this will include your full peer review and any attached files.

Reviewer #1: No

Reviewer #2: No

Reviewer #3: No

---

## [Decision Letter · Decision Letter 1]

4 Nov 2024

Dear Matt,

We are pleased to inform you that your manuscript entitled "Separable roles of the DNA damage response kinase Mec1ATR and its activator Rad24RAD17 during meiotic recombination" has been editorially accepted for publication in PLOS Genetics. Congratulations!

Yours sincerely,

Michael Lichten, Ph.D.

Academic Editor

PLOS Genetics

Eva Stukenbrock

Section Editor

PLOS Genetics

Aimée Dudley

Editor-in-Chief

PLOS Genetics

Anne Goriely

Editor-in-Chief

PLOS Genetics

Comments from the reviewers (if applicable):

Reviewer's Responses to Questions

**Comments to the Authors:**

Reviewer #1: The authors have appropriately addressed all remaining issues.

Reviewer #2: Reviewer concerns have been addressed.

Reviewer #3: The authors have responded adequately to my requests and comments.

**Have all data underlying the figures and results presented in the manuscript been provided?**

Reviewer #1: Yes

Reviewer #2: Yes

Reviewer #3: Yes

PLOS authors have the option to publish the peer review history of their article (what does this mean?). If published, this will include your full peer review and any attached files.

Reviewer #1: No

Reviewer #2: No

Reviewer #3: No

**Data Deposition**

http://datadryad.org/submit?journalID=pgenetics&manu=PGENETICS-D-24-00817R1

**Press Queries**

---

## [Editor Report · Acceptance letter]

2 Dec 2024

PGENETICS-D-24-00817R1 

Separable roles of the DNA damage response kinase Mec1ATR and its activator Rad24RAD17 during meiotic recombination 

Dear Dr Neale, 

We are pleased to inform you that your manuscript entitled "Separable roles of the DNA damage response kinase Mec1ATR and its activator Rad24RAD17 during meiotic recombination" has been formally accepted for publication in PLOS Genetics! Your manuscript is now with our production department and you will be notified of the publication date in due course.

With kind regards,

Anita Estes

PLOS Genetics

On behalf of:
